# Antimicrobial Susceptibility Testing Patterns of *Neisseria gonorrhoeae* from Patients Attending Sexually Transmitted Infections Clinics in Six Regions in Tanzania

**DOI:** 10.3390/tropicalmed7060089

**Published:** 2022-06-02

**Authors:** Said Aboud, Simon N. Buhalata, Onduru G. Onduru, Mercy G. Chiduo, Gideon P. Kwesigabo, Stephen E. Mshana, Alphaxard M. Manjurano, Mansuet M. Temu, Coleman Kishamawe, John M. Changalucha

**Affiliations:** 1Departments of Microbiology and Immunology, Epidemiology and Biostatistics, Muhimbili University of Health and Allied Sciences (MUHAS), Dar es Salaam P.O. Box 65001, Tanzania; buhalata@yahoo.com (S.N.B.); gpkwesigabo@gmail.com (G.P.K.); 2National Institute for Medical Research, Mwanza Research Centre, Mwanza P.O. Box 1462, Tanzania; amanjurano@yahoo.co.uk (A.M.M.); mmtemu@yahoo.co.uk (M.M.T.); kishamawe@yahoo.com (C.K.); john.changalucha@mitu.or.tz (J.M.C.); 3Department of Pathology, College of Medicine, Kamuzu University of Health Sciences, Blantyre P.O. Box 360, Malawi; ogyonduru@yahoo.com; 4National Institute for Medical Research, Tanga Research Centre, Tanga P.O. Box 5004, Tanzania; mercy_chiduo@yahoo.com; 5Department of Medical Microbiology, Catholic University of Health and Allied Sciences (CUHAS), Mwanza P.O. Box 1370, Tanzania; mshana72@yahoo.com

**Keywords:** antimicrobial resistance, quinolone-resistant *N. gonorrhoeae*, high level resistance, penicillinase producing-*neisseria gonorrhoeae*, multidrug resistant *N. gonorrhoeae*, extensively drug resistant *N. gonorrhoeae*

## Abstract

Antimicrobial resistance (AMR) is global health threat that is on the increase, and it has been adversely affecting the proper management of sexually transmitted infections (STI). Data on antimicrobial susceptibility testing patterns of *N. gonorrhoeae* are limited in local settings. We determined in vitro antimicrobial susceptibility and phenotypic profiles of *N. gonorrhoeae* isolated from STI/Outpatient Department (OPD) clinics. Minimum Inhibitory Concentrations (MIC) (µg/mL) were determined using E-Test and agar dilution methods for previously and currently recommended antimicrobial agents. A total of 164 *N. gonorrhoeae* isolates from urethral discharge and endocervical swabs were tested. The prevalence of resistant *N. gonorrhoeae* to tetracycline, norfloxacin, penicillin and ciprofloxacin were 98.6%, 82.2%, 84.3% and 75.6%, respectively. None of the isolates was resistant to kanamycin. Penicillinase producing *N. gonorrhoeae* (PPNG) was found to be 73.7%, with 56.7% and 43.3% observed among isolates from women and men, respectively. Tetracycline resistant-*N. gonorrhoeae* (TRNG) was found to be 34.0%, and QRNG with HLR to ciprofloxacin was 79.9%. The overall MDR-NG was 79.9%, and XDR-NG was 3.6%. MIC_50_ and MIC_90_ were 4.0 and 8.0 and 2.0 and 4.0 µg/mL for ciprofloxacin and norfloxacin, respectively. Dendrograms showed that 44 phenotypic groups are associated with a high rate of AMR among high MDR-NG and moderate XDR-NG isolates. The predominant groups of quinolone-resistant *N. gonorrhoeae* (QRNG)+PPNG (34.7%) and QRNG+PPNG+TRNG (32.9%) were observed among the isolates having HLR to ciprofloxacin. We reported a high prevalence of AMR (>90%) to previously recommended antimicrobials used for the treatment of gonorrhoea. Multidrug resistant *N. gonorrhoeae* (MDR-NG) was highly reported, and extensively drug resistant (XDR-NG) has gradually increased to the currently recommended cephalosporins including ceftriaxone and cefixime. Heterogeneous groups of QRNG+PPNG+ and QRNG+PPNG+TRNG were highly resistant to penicillin, tetracycline, ciprofloxacin and norfloxacin. A surveillance program is imperative in the country to curb the spread of AMR.

## 1. Introduction

Antimicrobial resistance (AMR) to *Neisseria gonorrhoeae* has been commonly observed against many antimicrobials including penicillins, tetracyclines, and fluoroquinolones [1,2,3]. Since the discovery of antimicrobials, *N. gonorrhoeae* strains have extremely developed high-resistance against many classes of antimicrobials [4]. Hence gonococcal infections globally are undoubtedly reported to be higher [3,5]. Antibiotic-resistant *N. gonorrhoeae* can evolve and spread rapidly [6]. Over 60% of *N. gonorrhoeae* cases have developed resistance to previously recommended antimicrobial agents [3,7,8]. The current data on AMR are incomplete and under-reported due to poor reporting systems, lack of clinical or laboratory diagnostic expertise and high rates of asymptomatic infections, especially in women [3,9]. There is, therefore, a need for enhanced global collaborative efforts that include better prevention, early diagnosis, management of partner and strengthened surveillance of *Neisseria gonorrhoeae* [10].

*N. gonorrhoeae* is a genetically diverse microorganism that can take up DNA at all stages of its life cycle, either from other gonococci, related pathogenic and non-pathogenic species or from bacteria of other genera [3,6]. This ability has made *N. gonorrhoeae* particularly efficient at acquiring resistance mechanisms to most classes of antimicrobial agents. Important resistance mechanisms include: pilus attachment to mucosal epithelium; enzymatic destruction of the antibiotics (penicillinase); target modification or protection (e.g., penicillin, cephalosporins); efflux of antimicrobial agents (most classes of antibiotic); and decreased influx of antimicrobial agents (e.g., penicillin, tetracycline). A mixture of resistance mechanisms usually present in a single gonococcal cell and/or a combination of genes coupled with many mutations within a specific gene is often required to achieve clinical levels of resistance to a particular antibiotic [3,6,10,11,12,13]. These mechanisms have made *N. gonorrhoeae* resistant to many antimicrobial agents such as fluoroquinolones and the currently recommended extended-spectrum cephalosporins (ESCs), which have undoubtedly helped it persist in the human populations [1,2,14]. High-level resistance [HLR] spreads rapidly through plasmid-mediated, while low-level resistance emerges through chromosomal-mediated and disseminates slowly [15,16].

Development of AMR to the previously recommended antimicrobials has been frequently reported worldwide [1,17]. The emergence of quinolone-resistant *N. gonorrhoeae* (QRNG) as a public health problem in Africa was first reported in surveys conducted between 2003 and 2006 in South Africa [18,19,20]. QRNG have been accelerated by fluoroquinolones used to treat other bacterial infections [21]. Recent studies conducted between 2005 and 2013 reported high rates (25–70%) of QRNG among different populations [1,22,23]. It is no doubt that antibiotic resistant-*N. gonorrhoeae* emerges in regions with a large informal health sector, and the use of common antibiotics is not well-controlled [10].

Further, penicillinase-producing *N. gonorrhoeae* (PPNG) was first reported in South Africa in 1977, and, in the 1980s, penicillin resistance reached proportions which necessitated its exclusion from use [24,25]. The WHO recommends that no antibiotic should be used when more than 5% of the isolates are resistant [4,5,21]. Previous data indicated that a high rate of PPNG (56%) was demonstrated in Kenyan isolates [26]. Some studies conducted between 1995 and 2007 observed high rates ranging from 26% to 60% in eastern and central African countries [27,28]. Tetracycline is among the superseded antibiotics, and the plasmid-mediated tetracycline resistant- *N. gonorrhoeae* (TRNG) is a high level that spreads rapidly, but tetracycline is still used in the informal health sectors, particularly in resource-constrained countries [27,28,29].

Recent reports of *N. gonorrhoeae* strains exhibiting potential resistance to ESCs have been documented [14,30,31]. There is a growing fear that if there are no concerted efforts to combat *N. gonorrhoeae*, it will become untreatable using antimicrobial monotherapy [3,14,31,32,33]. In most settings, ceftriaxone has been recommended as the only empirical first-line antimicrobial monotherapy for gonorrhoea [6,14,30], although World Health Organization (WHO) and Centers for Disease Control and Prevention (CDC) [6,17,34] have reported sporadic treatment failures. However, due to a diminished cache of drugs to treat gonorrhoea, most countries have recommended a combination of ceftriaxone and azithromycin in an attempt to ensure effective therapy and slow potential resistance to ESCs by reducing the likelihood that a *N. gonorrhoeae* isolate would survive a concomitant exposure to two antimicrobial agents with distinct mechanisms of action [4]. HLR to spectinomycin probably occurs by point mutations affecting their ribosomal target sites [10].

The quality data on AMR is sparse or unavailable. Most treatment guidelines, particularly in African countries, still recommend ciprofloxacin as the first-line therapy for the syndromic management of gonorrhoea and other treatable STIs [22,35]. Lack of microbiological laboratory capacity to perform qualitative and quantitative (MIC) in vitro susceptibility testing, limited use of culture-based diagnosis, which is essential for susceptibility testing and lack of surveillance programmes for monitoring gonococcal antimicrobial resistance, have been considered as important obstacles to obtaining quality data from in vitro susceptibility studies. A surveillance program of gonococcal AMR is imperative to detect resistant isolates and monitor trends of drug resistance to clinical isolates [3,26,36].

In Tanzania, ciprofloxacin was still recommended as the first-line treatment for gonorrhoea, although empirically, ceftriaxone is frequently used without formal prescriptions [35]. Previous data indicated a high and rapid increase of QRNG from 10% to >50% between 1995 and 2007 in Eastern African countries, including Tanzania [26,27,28,37]. However, there are no current data on antimicrobial susceptibility to guide syndromic management of gonorrhoea and other bacterial infections. Therefore, the present study was conducted to determine in vitro antimicrobial susceptibility testing and describe phenotypic antimicrobial profiles of *N. gonorrhoeae* isolates.

The study aimed to determine the prevalence of AMR of *N. gonorrhoeae* isolates to previously and currently recommended antimicrobials and characterize phenotypic characteristics based on the combination of antimicrobial resistance mechanisms. Data generated will be used to revise current treatment guidelines and inform policymakers for evidence-based decision making for the management and control of AMR of *N. gonorrhoeae* isolates in the country and globally.

## 2. Materials and Methods

### 2.1. Clinical Isolates

Clinical isolates of *N. gonorrhoeae* (N = 164) investigated in this study were collected consecutively from men with urethritis (urethral discharge and/or dysuria) and women with vaginal discharge or cervical infection recruited between 2014 and 2016. Eight health care facilities (STI/OPD clinics) were randomly selected from a total of 16 clinics in six regions of Dar es Salaam (1), Dodoma (1), Mwanza (2), Mbeya (2), Tanga (1) and Kilimanjaro (1). After obtaining written informed consent, participants were recruited and enrolled in each study site. However, written assent was obtained from a parent or a guardian for a patient aged 15 years old.

Cotton swabs used to obtain endocervical (n = 52) and urethral (n = 112) swab specimens were immediately streaked onto a modified Thayer-Martin (MTM plate), a selective medium with 1% Vitox and Vancomycin Colistin Amphotericin Trimethoprim (VCAT) supplements to isolate *N. gonorrhoeae* according to manufacturer’s instructions. Inoculated plates were incubated at 36 °C in 5% carbon dioxide for 24–48 h, and *N. gonorrhoeae* was identified by colonial morphology, Gram’s stain, and oxidase test. Further, *N. gonorrhoeae* isolates were confirmed by Real Time PCR assay. For preservation, gonococcal strains were subcultured onto plain MTM agar plates, incubated in similar conditions [38], harvested after a 24 h. incubation and preserved in duplicates in skimmed milk with 10% glycerol. The duplicates were stored at −20 °C freezer until the time for analysis.

### 2.2. In Vitro Antimicrobial Susceptibility Testing

#### 2.2.1. E-Test

Following confirmation of a gonococcal isolate, an in vitro susceptibility testing was performed using a confluent growth on a plain MTM. A standardized inoculum (equivalent to 0.5 McFarland or 1.0 × 10^8^ cfu/mL) of bacterial suspension was applied onto the surface of a diagnostic susceptibility testing (DST) plate to determine the quantitative minimum inhibitory concentration (MIC; µg/mL) to ciprofloxacin, ceftriaxone, cefixime, and azithromycin. Briefly, E-Test (BioMérieux AB, Solna, Sweden) was performed on a DST Medium Base (Oxoid, UK) supplemented with 1% Vitox (OXOID, Basingstoke, Hampshire, UK) according to manufacturer’s instructions. It is a strip with a graduated gradient of known concentrations of an antimicrobial agent, and it is calibrated to give a result in MIC values. The MIC_50_ and MIC_90_ is the amount of antibiotic which will inhibit 50% and 90% of strains in vitro tested [39]. A test-strip from each antibiotic was placed at the centre on the surface of a 90 mm plate, containing a nonselective medium inoculated with a standardized inoculum of bacterial suspension, with the highest concentration of the strip placed toward the edge of the plate.

#### 2.2.2. Agar Dilution Method

Gonococcal isolates were subcultured from frozen stocks onto an MTM plate, and the resulting colonies were re-subcultured onto a nonselective MTM plate. Briefly, a standardized inoculum of bacterial suspension (equivalent to 0.5 McFarland or 1.0 × 10^8^ cfu/mL) was also applied onto the surface of each agar plate containing serial dilutions of antimicrobial agents. The lowest concentration that completely inhibited the growth of the organism was recorded as the MIC value for each antibiotic. Inoculation was performed using a multipoint inoculator. Concentrations of antimicrobial agents used were: penicillin, 0.002–32 µg/mL; tetracycline, 0.008–32 µg/mL; norfloxacin, 0.008–32 µg/mL; spectinomycin, 1–256 µg/mL; erythromycin, 0.008–256 µg/mL; kanamycin, 1–256 µg/mL; and cefuroxime, 0.002–2 µg/mL.

For both methods, plates were incubated for 18–20 h at 36.5 °C in a 5% CO_2_—enriched atmosphere, based on the manufacturer’s instructions (Becton, Dickinson and Company, Franklin Lakes, NJ, USA). Results were read, and each MIC breakpoint was recorded in result form following the passing of reference controls (WHO controls) within a limit difference of 0 or ±1 serial dilution. MIC breakpoints with > or < MIC values were rounded to the next serial dilution, e.g., <0.016 was rounded to 0.008 and >32 was rounded to 64 µg/mL for convenience. Poor or no growth in the control plate was subcultured and repeated to avoid erroneous readings resulting from an isolate’s poor or slow growth. For each MIC breakpoint, a MIC µg/mL value was recorded, and results were interpreted as susceptible (S), intermediate susceptible (I) and resistant (R) according to the MIC breakpoints stated by CLSI criteria [40], except for azithromycin that used breakpoints recommended by the European Committee on Antimicrobial Susceptibility Testing (EUCAST) [41]. Criteria for decreased susceptibility to ceftriaxone (MIC ≥ 0.125 µg/mL) and cefixime (MIC ≥ 0.25 µg/mL) were defined by WHO [9].

### 2.3. Statistical Analysis

Data analysis was completed using Epi Info 7.2 CDC Atlanta Georgia USA and STATA 13.0 (Stata Corp, Timberlake, NC, USA). Prevalence for AMR was calculated from each antimicrobial agent tested. Values of MIC determined by serial dilutions/calibrated concentrations were presented as mean. MIC values of antimicrobial agents at which 50% and 90% of the isolates were inhibited were calculated and reported as MIC_50_ and MIC_90_, respectively, and the range of each antibiotic was reported. Logistic regression analysis was performed to determine the associations of AMR among the clusters of *N**. gonorrhoeae strains*.

Phenotypic clustering analysis for isolates was determined by constructing dendrograms and heat map analysis based on each antimicrobial resistance profile. A dendrogram analysis (Z-score and complete linkage) was performed in R software v.3.5.1 to characterize phenotypic clusters associated with AMR among NG strains. Coefficient values (Cophenetic coefficient 0.75 and delta 0.5 and 1.0) were used to evaluate the efficiency and validity of dendrograms and heatmap analysis. Clustering was completed based on in vitro susceptibility of an isolate to ciprofloxacin either susceptible denoted as non-quinolone resistant *N. gonorrhoeae* (NQRNG) or quinolone-resistant *N. gonorrhoeae* (QRNG) respectively. High-level resistance [HLR] and low-level resistance [LLR] were determined by MIC ≥ 4 µg/mL and MIC > 1 to <4 µg/mL to ciprofloxacin, respectively (CLSI, 2013). MDR was defined as a resistance of more than one of previously recommended antimicrobial agents namely penicillin, ciprofloxacin, norfloxacin, tetracycline, and erythromycin, plus one or more of the currently recommended antimicrobial agents—ceftriaxone, cefixime, cefuroxime, spectinomycin, and kanamycin and azithromycin. XDR was defined as a resistance of one or more antimicrobial agents to currently recommended regimen plus three or more of previously recommended antimicrobial agents (CLSI, 2013). Findings were presented as odds ratio and 95% CI. A *p*-value of ˂0.05 was regarded as statistically significant.

## 3. Results

### Prevalence of Antimicrobial and Multi-Drug Resistance

A total of 164 clinical isolates of *N. gonorrhoeae* were tested for in vitro antimicrobial susceptibility tests. Twenty-four isolates were not recovered during the in vitro susceptibility testing by the agar dilution method.

*N. gonorrhoeae* showed resistance to tetracycline 136 (97.1%), norfloxacin 118 (84.3%), penicillin 115 (82.2%) and ciprofloxacin 125 (75.6%). Resistance to the currently recommended antimicrobial agents, namely spectinomycin, azithromycin, cefuroxime, cefixime, and ceftriaxone, was 3.5%, 3.1%, 2.1%, 1.2% and 0.6%, respectively. *N. gonorrhoeae* showed intermediate resistance to kanamycin (2.1%) (Table 1).

Of 164 clinical isolates, 124 (75.6%) were collected from Mwanza, followed by 15 (9%) in Dar es Salaam and 9 (5.5%) in Tanga. However, the lowest rate of positive cultures of 3.25% was collected in Dodoma, Mbeya, and Kilimanjaro regions. For QRNG, 50.1% and 25.8% were isolates having HLR and LLR, respectively, in the Mwanza region. Similar rates of 76.6%, 10.5%, and 6.3% were found for PPNG and TRNG in Mwanza, Dar es Salaam, and Tanga, respectively. Resistance rates of 65.2%, 4.2%, and 3.0% for tetracycline were found in Mwanza, Dar es Salaam and Tanga, respectively. Ceftriaxone resistance of 1.2% was found in Mbeya and Tanga. Azithromycin resistance of 2.4% was found in Mwanza, whereas resistance for spectinomycin was found in Mwanza (2.7%) and Tanga (1.2%).

The overall mean MICs were 5.3 µg/mL and 5.1 µg/mL for ciprofloxacin and norfloxacin, respectively (Table 2). MIC50 and MIC90 of ciprofloxacin and norfloxacin to *N. gonorrhoeae* were 4.0 and 8.0, and 2.0 and 4.0 µg/mL, respectively.

The study observed the mean MICs of 0.023, 0.047, and 0.064 µg/mL of ceftriaxone, cefixime, and cefuroxime. However, MIC_50_ and MIC_90_ of these antibiotics were still reported at the lowest MIC break points, despite the fact that 0.6% to 3.5% of all isolates had AMR against cephalosporins and spectinomycin (Table 2). Between 3% and 3.6% of isolates had a mean MIC range of 0.032 to 0.75 µg/mL for ceftriaxone. One isolate was resistant to ceftriaxone at 1.5 µg/mL and cefixime at 2 µg/mL. Three isolates were resistant to cefixime; two at 2 µg/mL and one at 1 µg/mL. The same isolates were also resistant to cefuroxime at 2 µg/mL. Furthermore, two isolates were resistant to azithromycin; one at 256 µg/mL and another at 2 µg/mL and five isolates were resistant to spectinomycin; three at 256 µg/mL and two at 128 µg/mL.

The overall prevalence of PPNG was 73.7% (95% CI: 65.7, 80.8%), with 56.7% and 43.3% observed in women and men, respectively. The overall prevalence of QRNG was 87.8% and those having high-level resistance [HLR], low-level resistance [LLR] and susceptible (NQR) were 52.4% (86), 23.2% (38), 24.4% (40), respectively. Prevalence of TRNG, non-TRNG and tetracycline-susceptible were 89.3% (125), 9.3% (13) and 1.4% (2), respectively.

Clustered dendrograms showed 44 phenotypic groups, determined by the euclidean distance method with complete linkage when in vitro susceptibility testing results were clustered in a dendrogram and heatmap analysis (Figure 1). Of 44 groups, 17 consisted of a major cluster and had HLR to quinolone. Thirteen groups had LLR, to quinolone and the remaining fourteen had non-quinolone resistance (NQR-series) with MDR and XDR, respectively. The Q[H]-series that consisted of the predominant group accounted for 52.3% (83) of all isolates, followed by the Q[L]-series 23.2% (38) and NQR-series 24.4% (40) (Figure 1).

In a separate analysis of clustered dendrograms, in vitro susceptibility results were compared with phenotypic groups to determine the association of antimicrobial resistance among isolates. Three main clustered dendrograms were observed. Cluster 1 showed higher AMR than previous recommended antimicrobial agents with PPNG (Figure 2a). Cluster 2 had mixed patterns of phenotypic characteristics. Phenotypic characteristics among fluoroquinolone-susceptible and QRNG isolates are shown in Figure 2a,b, respectively. For instance, Figure 2b cluster 1 showed that all isolates were susceptible to ciprofloxacin, tetracycline, norfloxacin, penicillin, azithromycin, erythromycin, cefuroxime, ceftriaxone, cefixime, kanamycin and spectinomycin. Cluster 2 isolates were all resistant to all antimicrobial agents used except penicillin-sensitive isolates used as an out-group (Figure 2c). Three main clusters with mixed phenotypic characteristic patterns are shown in Figure 2c.

## 4. Discussion

The present study has observed a high prevalence of previously recommended antimicrobial agents, namely penicillin, tetracycline, ciprofloxacin and norfloxacin. Many previous studies have reported treatment failure for gonorrhoea by ciprofloxacin, even though, in resource-limited countries, ciprofloxacin is still used to treat gonorrhoea and other infections, especially in informal health sectors. HLR to ciprofloxacin associated with MDR among *N. gonorrhoeae* isolates as classified by the current definition was also observed in this study. In vitro resistance for MDR-NG and XDR-NG was <4% to the currently recommended antimicrobial agents namely cephalosporins and spectinomycin. This in vitro susceptibility study reports extensively drug resistant-*N. gonorrhoeae* (XDR-NG) of >3% to ceftriaxone, cefixime, cefuroxime and spectinomycin in Tanzania. In comparison, some studies [1,2] also reported MDR-NG using these definitions described previously in the current study. Recently, some studies reported >20% of *N. gonorrhoeae* isolates were MDR-NG having HLR to ciprofloxacin between 2000 and 2009, according to conventional definition [1]. In African countries, penicillin, tetracycline, and ciprofloxacin resistance was 65%, 97%, and 11%, as reported in Kisumu, Kenya, from 2002 to 2009 [26]. However, in 2007, a study conducted in Malawi showed high levels of resistance to tetracycline (TRNG) and penicillin, but all isolates were susceptible to ciprofloxacin [4]. An increase of >50% was reported in three regions of Kenya in 2012, a few years later [5]. In Tanzania, PPNG was reported in 2009 at a low rate [6].

Five percent of *N gonorrhoeae* strains had been reported to have a decreased susceptibility (but no resistance) to currently recommended ESCs [7]. Most reports have documented both in vitro and molecular resistance to cephalosporins and azithromycin based on a case by case studies and some cross-sectional observations [1,7,8,9,10,11,12,13,14]. Though sporadic cases of HLR to ESCs have been reported in France and other parts of the world [12,13,14,15,16], many African countries are unaware of the true prevalence of AMR to cephalosporins as in vitro susceptibility to these agents may not be regularly monitored [17]. In Tanzania, cephalosporins such as ceftriaxone, cefixime, cefuroxime, and macrolides (azithromycin and erythromycin) are frequently used in formal and informal health sectors and sometimes are found in over the counter (OTC) shops. In addition, the emergence of in vitro resistance of ceftriaxone, cefixime and other cephalosporins to *N. gonorrhoeae* has not been established [13,14], but only molecular resistance has been documented in recent studies [18,19,20].

The MIC_50_ and MIC_90_ are useful parameters for monitoring trends of MIC values, of which the increase suggests a gradual or rapid development of resistance to antimicrobial agents. In the current study, MIC_50_ and MIC_90_ were highly increased (>5.0 µg/mL) to penicillin, tetracycline, ciprofloxacin, and norfloxacin, suggesting that these antimicrobial agents are no longer recommended for the treatment of gonorrhoea. However, a slight increase (but still susceptible) in MIC_50_ and MIC_90_ was observed to ceftriaxone and cefixime, suggesting a slight paradigm shift where *N. gonorrhoeae* may undergo resistance to these agents in the near future, particularly in high-risk populations. Our findings are comparable to those documented in previous and current studies. A study conducted in Kenya between 2002 and 2009 revealed that MIC50 and MIC90 were highly increased (>5.0 µg/mL) to penicillin and tetracycline, respectively [40], indicating that these antibiotics were no longer effective against *N. gonorrhoeae*. However, a mean MIC from 0.004 to 4.0 µg/mL for both MIC50 and MIC90 to ciprofloxacin was observed between 2007 and 2009, indicating that ciprofloxacin was ineffective for treating gonorrhoea [3].

In vitro Susceptibility testing patterns and clustered dendrograms showed that phenotypic groups were diverse and heterogeneous. The predominant phenotypic groups of QRNG+PPNG+ and QRNG+PPNG+TRNG+, respectively, were observed to be highly increased to >30% among the *N. gonorrhoeae* isolates with HLR to ciprofloxacin observed in the study. Therefore, the current report highlights that heterogeneous phenotypic groups of QRNG+PPNG+ and QRNG+PPNG+TRNG+ were predominant and thus suggest a selective pressure from antibiotic use. This is a strong driver of the potential emergency of AMR to currently recommended cephalosporins. A previous study conducted in 2009–2013 showed >10% of predominant groups of QRNG+PPNG+ and QRNG+PPNG+TRNG+ were also observed among the isolates with HLR to ciprofloxacin [1]. A high prevalence of 42.7% for tetracycline (TRNG) was observed among the isolates with HLR to ciprofloxacin, although some previous studies suggested [1,3,7] returning to ‘older’ antibiotics may be another viable option for gonorrhoea treatment. Tetracycline is no longer used in most resource-constrained countries for over a decade [41]. However, current reports demonstrated a high prevalence of resistant *N. gonorrhoeae* isolates to tetracycline [9,10,11].

There is an even higher rate of treatment failure for fluoroquinolones at the currently accepted breakpoint, which is MIC ≥1 µg/mL for ciprofloxacin. Treatment failures occur in about 60% of patients infected with gonococcal strains for which the MIC is ≥ 1 µg/mL [26]. Therefore, the mean MICs to ciprofloxacin was highly increased and thus indicating treatment failure to this antibiotic for 100% among these isolates. In recent years, some studies observed molecular resistance of *N. gonorrhoeae* isolates to ceftriaxone correlated to 1 µg/mL (WHO “O” reference strain) [42]. The current study has also demonstrated an in vitro resistance to ceftriaxone and cefixime. The study findings are in agreement with (included WHO reference strains) previously documented studies [9,11,12,13,14,21].

The emergence of antibiotic-resistant *N. gonorrhoeae* can occur silently and spread rapidly without ongoing microbiological surveillance [21,22]. A recent study challenges the currently prevailing idea of screening and treating infections to limit the spread of *N. gonorrhoeae* [17]. Increased treatment rates will ultimately result in a faster spread of AMR. Factors influencing and contributing to the spread of AMR include but are not limited to the irrational use of antibiotics and the introduction of resistant strains by international travelers. The dominant gonococcal strains and their antibiotic profiles can change rapidly, and therefore short-term studies need to be repeated regularly to guide treatment regimens [23].

## 5. Conclusions and Recommendations

MDR-NG was found to be high and XDR-NG gradually increased to the currently recommended cephalosporins including ceftriaxone and cefixime. Heterogeneous groups of QRNG+PPNG+ and QRNG+PPNG+TRNG were observed to be highly resistant to penicillin, tetracycline, ciprofloxacin and norfloxacin. Ceftriaxone and cefixime are still effective in treating gonorrhea, but effective measures must be taken to mitigate the potential growing of XDR-NG strains. A surveillance program is imperative in the country to curb the spread of AMR.

## 6. Strengths and Limitations

The high prevalence of AMR patterns observed from the clinical isolates obtained from patients attending STI/OPD clinics in six regions may not be generalizable to entire population. However, our findings remain striking that, the high prevalence of AMR to the previously and currently recommended antimicrobials among the isolates with high-level resistance to fluoroquinolones is a strong indicator that resistance of *N. gonorrhoeae* isolates may be common in STI clinics and other populations. The current study’s strength is the facts that standards and WHO recommended quantitative MIC methods of agar dilution and E-test platforms were used to determine in vitro susceptibility to previously and currently recommended antimicrobial agents. Further, WHO reference controls were also included during testing and analysis, and the results were comparable [26].

## Figures and Tables

**Figure 1 tropicalmed-07-00089-f001:**
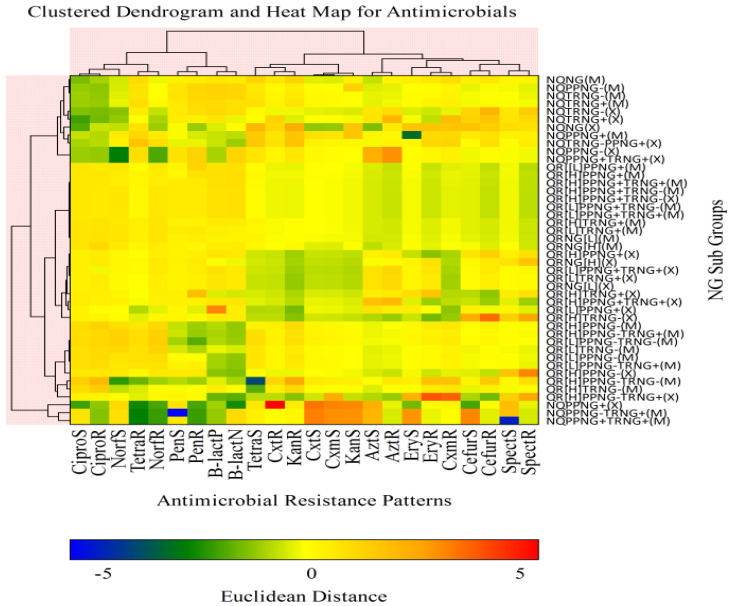
Clustered dendrogram with heatmap analysis, showing the relationship of phenotypic groups associated with AMR among *N. gonorrhoeae* isolates as determined by Euclidean Distance method. Key: + = positive, (e.g., PPNG+ = Penicillinase producing-*N. gonorrhoeae* (βeta-lactamase+)), − = negative (e.g., PPNG− = non penicillinase producing-*N. gonorrhoeae* (βeta-lactamase-ve)), QR = Quinolone resistance (fluoroquinolone resistance of ciprofloxacin and or norfloxacin resistance), [H] = high level resistance (HLR), NG = *N. gonorrhoeae*, TRNG+ = plasmid mediated tetracycline resistant-*N. gonorrhoeae* (MIC ≥ 16 µg/mL), NQ = non quinolone resistance (ciprofloxacin susceptible), [L] = low level resistance (LLR) (MIC < 4 to ≥1 µg/mL to ciprofloxacin), TRNG− = non-plasmid mediated tetracycline resistant-*N. gonorrhoeae* (MIC < 16 µg/mL), M = MDR; Multidrug Resistance, X = XDR; Extensively Drug Resistance, S = susceptible, R = resistant, Cipro = ciprofloxacin, Norf = norfloxacin, Azt = azithromycin, Pen = penicillin, Tetra = tetracycline, Cxt = ceftriaxone, Cxm = cefixime, Cefur = cefuroxime, Spect = spectinomycin, B.lact = βeta-lactamase, N = negative, P = positive, Euclidean Distance (>−5 to >5) = Z-scores (BLUE to RED) were calculated from susceptible to resistant-*N. gonorrhoeae* isolates, respectively.

**Figure 2 tropicalmed-07-00089-f002:**
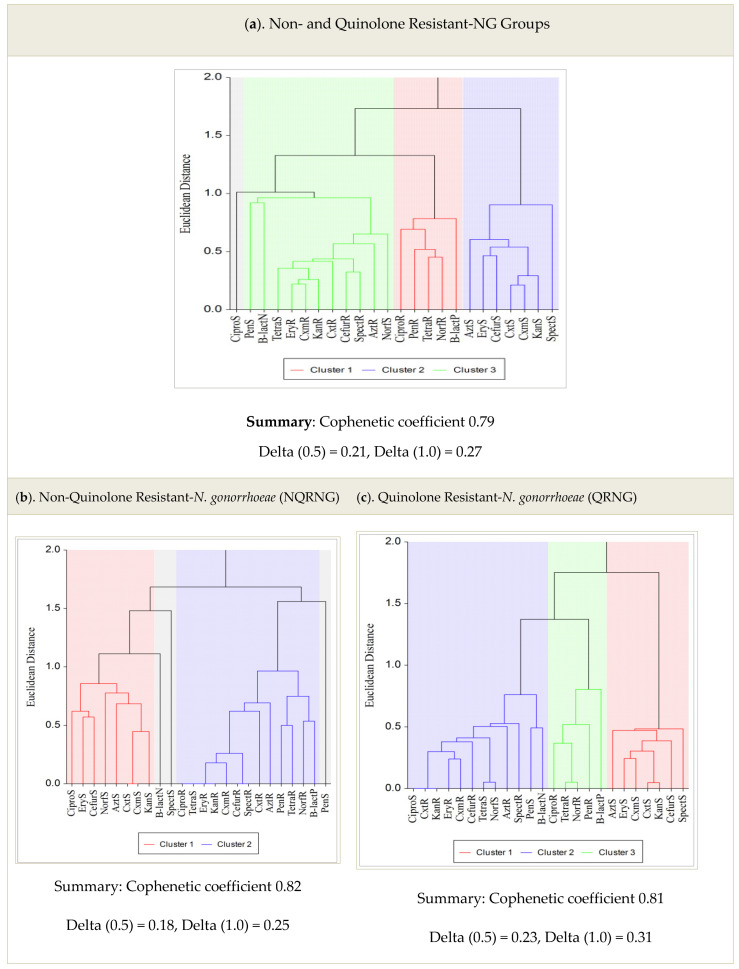
(**a**–**c**): Clustered Dendrograms and Phenotypes among NQRNG and QRNG isolates.

**Table 1 tropicalmed-07-00089-t001:** Antimicrobial Susceptibility Testing Patterns of *N. gonorrhoeae* Isolates (N = 164) Collected From Six Regions in Tanzania.

Antimicrobial	Susceptible n (%)	Intermediate n (%)	Resistant n (%)
Ciprofloxacin	20 (12.2)	19 (11.5)	125 (75.6)
Norfloxacin	11 (7.8)	14 (10.0)	118 (84.3)
Tetracycline	2 (1.4)	1/140 (0.7)	136 (97.1)
Penicillin	9 (6. 4)	16 (11.4)	115 (82.2)
Ceftriaxone	161 (98.2)	1 (0.6)	1 (0.6)
Cefixime	160 (97.6)	2 (1.2)	2 (1.2)
Cefuroxime	136 (97.1)	2 (0.7)	3 (2.1)
Azithromycin	158 (96.3)	2 (1.2)	5 (3.1)
Erythromycin	120 (85.7)	18 (12.8)	2 (1.4)
Spectinomycin	135 (96.4)	0 (0.0)	5 (3.5)
Kanamycin	137 (92.9)	3 (2.1)	0 (0.0)

**Table 2 tropicalmed-07-00089-t002:** Mean MICs, MIC_50_ and MIC_90_ for Previously and Currently Recommended Antimicrobial Agents Commonly Used to Treat *N. gonorrhoeae*.

Antimicrobial	MIC (µg/mL), Men (n = 112)			MIC (µg/mL), Women (n = 52)			Overall MIC (µg/mL)			
	Mean (SD)	MIC_50_	MIC_90_	Mean (SD)	MIC_50_	MIC_90_	Mean (SD)	MIC_50_	MIC_90_	Range
Ciprofloxacin	5.1 (5.5)	4.0	8.0	5.6 (7.3)	4.0	8.0	5.3 (6.5)	4.0	8.0	0.002–32
Norfloxacin	5.4 (9.6)	2.0	4.0	4.8 (7.3)	2.0	4.0	5.1 (8.1)	2.0	4.0	0.002–32
Tetracycline	27.2 (9.9)	32.0	32.0	27.7 (9.3)	32.0	32.0	27.5 (10.4)	32.0	32.0	0.016–32
Penicillin	21.1 (14.8)	32.0	32.0	20.9 (14)	32.0	32.0	21 (14.5)	32.0	32.0	0.004–32
Ceftriaxone	0.009 (0.03)	0.002	0.004	0.037 (0.17)	0.002	0.004	0.023 (0.13)	0.002	0.004	0.002–1.5
Cefixime	0.022 (0.058)	0.016	0.016	0.067 (0.241)	0.016	0.016	0.047 (0.18)	0.016	0.016	0.016–2
Cefuroxime	0.005 (0.004)	0.002	0.008	0.11 (0.43)	0.008	0.008	0.064 (0.32)	0.008	0.008	0.002–2
Azithromycin	3.9 (30.7)	0.125	0.25	0.24 (0.27)	0.125	0.19	1.9 (19.6)	0.19	0.25	0.002–256
Erythromycin	0.08 (0.15)	0.032	0.063	0.23 (0.57)	0.063	0.25	0.18 (0.44)	0.063	0.125	0.002–4
Spectinomycin	17.8 (11.1)	16	32	35.5 (54.7)	16	32	29.8 (46.9)	16	32	1–256
Kanamycin	7.3 (6.5)	8.0	8.0	10.6 (13.9)	8.0	16	8.9 (11.2)	8.0	8.0	0.25–64

## Data Availability

We collected demographic and clinical information from all participants; these raw data were generated and analyzed in the Data Management Unit (DMU) at NIMR Mwanza centre. Datasets of this study are available and deposited in the local public domain of NIMR data server. The generated datasets will be made available to the editor on request from the corresponding author or Director NIMR Mwanza centre.

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
