# Peer review of "Antimicrobial Susceptibility Testing Patterns of Neisseria gonorrhoeae from Patients Attending Sexually Transmitted Infections Clinics in Six Regions in Tanzania"

_tropicalmed, 2022, doi:10.3390/tropicalmed7060089_

Round 1

Author Response

MUHIMBILI UNIVERSITY OF HEALTH AND ALLIED SCIENCES

SCHOOL OF MEDICINE

       DEPARTMENT OF MICROBIOLOGY AND IMMUNOLOGY

Ref No. PF/9071/1                                                                                                                15/05/2022

Editor,

MDPI

Tropical Medicine Editorial Office

St Alban-Anlage 66, 4052 Basel

Switzerland

Re: Submission of Responses to Reviewer’s Comments on the Manuscript Titled

Antimicrobial Susceptibility Pattern and Phenotypic Analysis of Neisseria gonorrhoeae from Patients Attending Sexually Transmitted Infections Clinics in Six Regions in Tanzania” by Aboud et al 

We would like to thank reviewer 1 for very good comments and suggestions that has helped to improve the manuscript. Below are point-to-point responses to reviewer’s comments. Changes made in the revised manuscript are highlighted in yellow.

Responses to Reviewer 1 Comments

Point 1:

Rephrase the title, the title is very long and confusing.

Response 1: We concur with reviewer’s comments, the title has been rephrased to read “Antimicrobial Susceptibility Testing Patterns of Neisseria gonorrhoeae from Patients Attending Sexually Transmitted Infections Clinics in Six Regions in Tanzania”

Point 2:

Introduction: The authors should elaborate in a few sentences on the pathology of N. gonorrhoeae

Avoid repetition; rather give more examples for area of use. Describe, relate, discuss (-> also for discussion)

Response 2: Details of mechanisms of AMR and examples are described in Introduction section, paragraph 2, line 61-73. 

Point 3:

Line 47: Please have a look again at this sentence__ The current data on AMR are incomplete and under-reported due to poor reporting systems, lack of clinical or laboratory diagnostic expertise and high rates of asymptomatic infections, especially in women [3,9]. However, the available data, usually in resource-poor settings suggest that resistance to these agents sometimes reaches unacceptable levels [10]. *This reference is quite old, its 1997 published article, why not authors search more recent articles to obtain the results.

Although you should include this in your introduction/discussion/conclusion how future research can answer those knowledge gaps. if possible, provide more references to defend this.

Response 3: Current reference on AMR in Neisseria gonorrhoeae by Wi T et al 2017 has been added to replace the 1997 reference. The sentence in paragraph 1 in the introduction section, line 56-58, has been recasted

Point 4

Line 52: N. gonorrhoeae is a genetically diverse microorganism which is able to take up DNA at all stages of its life cycle, either from other gonococci, related pathogenic and non pathogenic species, or from bacteria of other genera. Provide the references, and also complete the sentences with some examples.

Response 4: Reference number 3 and 6 have been added. Examples are provided from line 61-74.

Point 5

Line 52: Why not mentioned the other virulence factors, e.g; Pilus: its aids in attachment to mucosal epithelium, contains constant and hypervariable regions-analogous to immunoglobins-that contribute to antigenic diversity in gonococci, of course it leads to AMR.

Response 5: Reviewer’s comment has been incorporated in line 63

Point 5

Line 77: Further, penicillinase-producing N. gonorrhoeae (PPNG) was first reported in South 77 Africa in 1977 *Wrong reference quoted__Pham-Kanter GB, Steinberg MH, Ballard RC. Sexually transmitted diseases in South Africa. Genitourin Med. 1996;72(3):160–71.

Correct your reference_

Robins-Brown RM, Gaillard MRC, Koornhof HJ, Mauff AC. Penicillinase-producing Neisseria gonorrhoeae [leter].SAfrMedJ 1977;51:568.

Hallet AF, Appelbaum PC, Cooper R, Mokgokong S, Monale D. Penicillinase-producing Neisseria gonorrhoeae from South Africa [leter]. Lancet 1977;1:1205.

Response 5: Reviewer’s comments have been incorporating by adding two correct references cited above (Reference number 24 and 25)

Point 6

- Since patients with gonorrhea may also infected with chlamydia, treatment regime often includes a combination of agent targeting both organisms, please discus this information as well for resistivity development.

Response 6: It is true that patients with gonorrhea may also infected with chlamydia, however, the primary aim of the paper was to perform phenotypic analysis of Neisseria gonorrhoeae  

Point 7

- Do the prophylactic drugs for Ophthalmia neonatorum or suspected mother causes resistivity?

Response 7: Yes, there is a possibility to develop resistance if the suspected mother uses antimicrobial drugs without evidence of laboratory results. 

Materials and Methods

2.1. Clinical Isolates

Clinical isolates of N. gonorrhoeae (N=164) investigated in the study were collected consecutively from men with urethritis (urethral discharge and/or dysuria) and women with vaginal discharge or cervical infection

Point 8

*Although a purulent discharge is characteristics of local gonococcal infection, it is not possible to distinguished reliably between gonococcal discharge and that caused by other pathogens such as Chalmydia trachomatis on clinical examination. Did you check other pathogenic agents. 

Response 8: The primary aim of the current study is to present AST patterns of N. gonorrhoeae isolated, other pathogenic agents such as Chalmydia trachomatis, M genitalium and C albicans were also checked, however, they are not the primary focus of this report.   

Results

Point 9

Clinical isolates of N. gonorrhoeae from both male and female, but data represent in the results section not separated gender wise, if you have separate data for male and female for antibiotic susceptibility etc, please provide in supplementary file. As we know that this bacterium is a human pathogen and does not cause infection to other animals, therefore its reservoir is human, and transmission is direct. It showed that 50% women has a chance of becoming infected after a single intercourse with infected man, while in male its only 20%. Keeping this fact in mind, please provide the data.

Response 9: Number of clinical isolates of N. gonorrhoeae by gender is now presented in the results section.
Point 10

- The quality of all the images must be improved for better clarity and representations. For example, the Figure 1. Clustered dendrogram image are not clear at for the readers.

Response 9: Reviewer’s comments have been incorporated

Point 11

In the short conclusion section, aspects are presented that rather belong to the discussion than solely to the conclusion, as they have not been described/defined before.

Response 11: Conclusion has been recasted to address reviewer’s comments

Point 12

- Many typographical errors are present.

The above list is just meant to be a suggestion for improvement of the manuscript.

Response 12: Typographical errors have been corrected

We look forward to hearing from you soon

Said Aboud

Reviewer 2 Report

This determined in vitro antimicrobial susceptibility and phenotypic antimicrobial profiles 18 of N. gonorrhoeae isolated from Sexually Transmitted Infections/Outpatient clinics in six regions of Tanzania. Results from this study showed a high resistance of the isolates to four common antibiotics (tetracycline, norfloxacin, penicillin and ciprofloxacin) indicating the need to control antimicrobial resistance (AMR) in the country. This work is original and has the capability to add to knowledge in the field of AMR.

Title

The title, as presented, captures adequately the central theme of the study. However, the authors should italicize “Neisseria gonorrhoeae” in the title.

Abstract

The abstract, though written to capture the objective, methods, results, and conclusion of the research lacked a good introduction/background. Furthermore, abbreviations such as PPNG, TRNG, QRNG, HLR, MDR-NG, XDR-NG, MIC, and AMR were used in the abstract but were not defined at first mention. The presentation of the abstract should be improved upon.

Other comments on the abstract

Line 21: “WHICH TYPE OF SPECIMENS” appears vague. What does this denote?

Line 22: Add “The” before “Prevalence” and a comma after “penicillin”

Line 24: Add a comma after “73.7%” before “with”.

Line 25: Add a comma between “34%” and “and”. Also ensure that “34%” is presented in the same decimal place as other presented percentages.

Line 26: Add a comma between “79.9%” and “and”.

Line 27: Include “that” after “showed”.

Line 28: Replace the comma before “associated” with “are” and include “a” before “high”.

Line 30: Include “a” before “high prevalence”.

Line 31: Include “the” before “treatment”.

Line 32: include a comma before “and XDR-NG”.

Lines 31 - 36: For a better clarity, authors should modify these sentences “MDR-NG was highly reported and XDR-NG was found gradually increasing to the currently recommended cephalosporins including ceftriaxone and cefixime. Heterogeneous groups of QRNG+PPNG+ and QRNG+PPNG+TRNG were highly observed resistant to penicillin, tetracycline, ciprofloxacin and or norfloxacin. Surveillance program is imperative in the country in order to curb the spread of the AMR.” to “MDR-NG was highly reported, and XDR-NG gradually increased to the currently recommended cephalosporins, including ceftriaxone and cefixime. Heterogeneous groups of QRNG+PPNG+ and QRNG+PPNG+TRNG were highly resistant to penicillin, tetracycline, ciprofloxacin, and norfloxacin. A surveillance program is imperative in the country to curb the spread of AMR.”

Keywords

Are abbreviations allowed in the keywords?

Introduction

The introduction, presented, is comprehensive. It addresses the import of this research. The actual problem necessitating the conduct of this research was presented and well-justified. The aim of the study was also presented. However, the introduction is flawed with several grammatical, spelling, punctuation, and presentation errors that must be fix. Some of these have been listed below.

Line 42: For a more concise sentence, please delete “classes of”, add a comma before “including” and delete the semicolon after “including”. Also add a comma after “tetracyclines”.

Line 44: Replace the comma after [4] with a full stop and begin the next sentence with “Hence”.

Line 45: Add “to be” between “reported” and “higher”.

Line 50: Add a comma after “settings”

Line 52: Please change “which is able to” to “that can”

Lines 53-54: Add a hyphen between “non” and “pathogenic” and delete the comma after “species”

Line 58: Add “s” to “antibiotic” and remove the comma after “and”

Line 60: Add a comma after “cell”

Line 63: Replace “against” with “to”

Lines 62-65: Re-write this “These mechanisms have made N. gonorrhoeae to become resistant against many classes of antimicrobial agents such as fluoroquinolones and the current recommended extended spectrum cephalosporins (ESCs) with which have undoubtedly helped it persist in the human populations [1,2,14].” sentence as “These mechanisms have made N. gonorrhoeae resistant to many antimicrobial agents such as fluoroquinolones and the current recommended extended-spectrum cephalosporins (ESCs), which have undoubtedly helped it persist in the human populations [1,2,14].”

Line 66: Please delete “emerges and” from the sentence and add a hyphen between “plasmid” and “mediated”

Line 68: Replace “previous” with “previously”

Line 75: Replace “where there is” with “with” and add a comma after “sector”

Line 78: Add a comma after “1977” and “1980s”. Also insert “the” before “1980s”

Line 80: Add “a” before “high”

Line 81: Add a comma after “2007”

Line 83: Add a comma after “antibiotics”

Line 85: Add a comma after “sectors”

Line 87: Begin the sentence with “Recent reports” and delete “an emergence of”

Line 88: Replace “extended spectrum cephalosporins (ESCs)” with “ESCs” since it has been defined earlier.

Lines 90-93: Re-write this sentence “In most settings, ceftriaxone has been recommended as the only empirical first-line antimicrobial monotherapy for gonorrhoea [6,14,29], despite the fact that, sporadic treatment failures to ceftriaxone have been reported by WHO and CDC [6,17,33].” as “In most settings, ceftriaxone has been recommended as the only empirical first-line antimicrobial monotherapy for gonorrhoea [6,14,29], although WHO and CDC [6,17,33] have reported sporadic treatment failures with ceftriaxone.”

Line 93: Write out “WHO” and “CDC” in full.

Line 100-102: Re-write this sentence “The quality data on antimicrobial resistance is sparse or unavailable, and most treatment guidelines particularly in African countries still recommend ciprofloxacin as the first line therapy for syndromic management of gonorrhoea and other treatable STIs [22,34].” as ‘’The quality data on antimicrobial resistance is sparse or unavailable. Most treatment guidelines, particularly in African countries, still recommend ciprofloxacin as the first-line therapy for the syndromic management of gonorrhoea and other treatable STIs [22,34].”

Line 110: Re-write “first line” as “first-line” and include a comma after “gonorrhoea”

Line 112: Delete “there was”

Line 113: Replace the semi-colon after “countries” with a comma

Lines 118-123: Re-write these sentences “The study aimed to determine the prevalence of AMR of N. gonorrhoeae isolates to previous and current recommended antimicrobials and characterization of phenotypic characteristics based on the combination of antimicrobial resistance mechanisms. Data generated will be used to revise current treatment guidelines and to inform policy makers for evidence based decision making for management and control of AMR of N. gonorrhoeae isolates in the country and globally.” as “The study aimed to determine the prevalence of AMR of N. gonorrhoeae isolates to previously and currently recommended antimicrobials and characterize phenotypic characteristics based on the combination of antimicrobial resistance mechanisms. Data generated will be used to revise current treatment guidelines and inform policymakers for evidence-based decision making for the management and control of AMR of N. gonorrhoeae isolates in the country and globally.”

Materials and methods

Line 126: Change “in the study” to “in this study”

Line 129: Write STI/OPD in full at first mention

Line 130: Add a comma after “Mbeya”

Lines 129-130: How many STI/OPD were randomly selected from a total of how many clinics in each or from the six region(s)?

Line 131-132: For clarity, re-write this sentence “Participants were recruited and enrolled in each study site after obtaining written informed consents” as “After obtaining written informed consent, participants were recruited and enrolled in each study site.”

Line 133: For conciseness, replace “whose age was” with “aged” and state the exact age of the patient (not < 16 yrs old) since it’s just one patient.

Lines 134-141: The isolation of N. gonorrhoeae was carried out according to which protocol? Authors should cite the standard/published protocol used in the isolation

Line 135: Add a comma after “(MTM plate)”

Line 136: “VCAT” was not defined

Line 139: Re-write this phrase “a plain MTM agar plates” as “plain MTM agar plates”

Line 141: Why have the authors decided to store the duplicates of the isolates at two different temperatures: -80 and -150 °C?

Line 143: The E-Test was carried out according to which protocol?

Line 147: Add “a” before “DST” and “the” before “quantitative”. What is DST? This abbreviation has not been defined.

Line 150: Add “a” before “graduated”

Line 151: Include a comma after “agent”

Line 154: Re-write “a 90-mm plate,” as “a 90 mm plate”

Line 158: Add a comma after “plate”

Line 162: Add “the” before “growth”

Line 168: Change “as per” to “based on” and add “the” before “manufacturer’s”. Add a comma after “read”.

Line 169: Add “the” before “passing”

Line 171: Add a comma after “dilution”

Line 172-174: Replace this sentence “Poor or no growth in the control plate was resubcultured and repeated to avoid erroneous reading which can be a result of poor or slow growth of an isolate.” with “Poor or no growth in the control plate was subcultured and repeated to avoid erroneous readings resulting from an isolate's poor or slow growth.”

Line 174: Add a comma after “recorded”

Line 182: Delete “individual”

Line 185: Add “were” before “inhibited”

Line 187: “Strains” should not be italicized

Line 188: Replace “the construction of” with “constructing”

Line 195: Add a hyphen between quinolone and resistant

Line 193-195: Please re-write this sentence “. Clustering was done based on in vitro susceptibility of an isolate to ciprofloxacin either susceptible denoted as non-quinolone resistant N. gonorrhoeae (NQRNG) or quinolone resistant N. gonorrhoeae (QRNG) respectively” to make better meaning.

Line 199: Add a comma after “tetracycline”

Line 200: Add “the” before “currently”

Line 204: Include a hyphen between “p” and “value”. Also, the general rule is to consider significant values at p-values less than 0.05 and not less or equal to 0.05. The authors should revise their results to ensure that p-values of 0.05 were not taken as significant.

Line 209: Add “the” before “agar”

Lines 212-213: Add a comma after “agents”, “cefixime’, “ceftriaxone”, and “1.2”. Also include “%” to “2.1” and “1.2”. Change the “were” before “3.5%” to “was”

Line 215: Add a comma after “Mwanza”

Line 217: Delete “respectively” and add a comma after “Mbeya”

Line 218: Add “the” before “Mwanza region”

Line 219: Add a comma after “10.5%” and delete “both” before “PPNG”

Line 220-221: Add a comma after “4.2%” and “Dar es Salaam”

Line 222: Delete “both” before “Mbeya”

Line 223: Add a comma after “Mwanza”

Lines 226-227: Fix the punctuation errors in this sentence “MIC50 and MIC90 of ciprofloxacin and norfloxacin to N. gonorrhoeae were 4.0 and 8.0 and, 2.0 and 4.0 µg/mL respectively” by re-writing it as “MIC50 and MIC90 of ciprofloxacin and norfloxacin to N. gonorrhoeae were 4.0 and 8.0, and 2.0 and 4.0 µg/mL, respectively.”

Line 227-228: For better clarity, re-write this sentence “Mean MICs of 0.023, 0.047 and 0.064µg/mL of ceftriaxone, cefixime and cefuroxime were observed in the study” as “The study observed the mean MICs of 0.023, 0.047, and 0.064µg/mL of ceftriaxone, cefixime, and cefuroxime.”

Line 231: Add “a” before “mean”

Line 237: Begin the sentence with “The”

Line 238: Add a comma before “with”

Line 240: Add a comma after “(38)”

Line 243: Add a comma after “groups” and delete “which were”

Line 244: Add “the” before “euclidean”

Line 245: Add “a” before “dendogram”

Line 246: Add “and” before “had” and delete the comma after “cluster”. Add a comma after “LLR to quinolone”

Line 249: Add a comma after “isolates” and include “the” after “by”

Line 253-254: Rephrase “high AMR to” to “higher AMR than”

Line 258: Insert a hyphen between “penicillin” and “sensitive”

Line 259: Delete “that were”

Discussion

The results were well discussed. However, the authors must fix all grammatical, punctuation, spelling, and presentation errors noticed.

Line 281: Add “a” before “high”

Line 282: Add a comma after “agents”

Lines 282-285: Modify this sentence “Treatment failure for gonorrhoea by ciprofloxacin has been reported by many previous studies, despite the fact that, in resource limited countries, ciprofloxacin is still used to treat gonorrhoea and other infections, especially in informal health sectors.” as “Many previous studies have reported treatment failure for gonorrhoea by ciprofloxacin, and even though, in resource-limited countries, ciprofloxacin is still used to treat gonorrhoea and other infections, especially in informal health sectors.”

Line 287: Delete “both” after “for”

Line 291: Delete “as” after “definitions”

Lines 294-296: Re-present this sentence “In African countries, penicillin resistance was 65%, tetracycline 97%, and ciprofloxacin 11% as reported in Kisumu, Kenya from 2002 and 2009[3].” as “In African countries, penicillin, tetracycline, and ciprofloxacin resistance was 65%, 97%, and 11%, as reported in Kisumu, Kenya, from 2002 to 2009[3].”

Line 298: Change the “The” at the beginning of the line to “An”, add a comma after “2012” and delete the “in” after “2012”

Line 302: Delete the “resistance” after “vitro”

Line 303: Add “a” before “case by case” and insert a hyphen between “cross” and “sectional”

Line 306-309: Re-write this sentence “In Tanzania cephalosporins such as ceftriaxone, cefixime and cefuroxime and macrolides (azithromycin and erythromycin) are frequently used both in formal and informal health sectors and sometimes are found in over the counter (OTC) shops” as “In Tanzania, cephalosporins such as ceftriaxone, cefixime, cefuroxime, and macrolides (azithromycin and erythromycin) are frequently used in formal and informal health sectors and sometimes are found in over the counter (OTC) shops.”

Line 311: Add a comma after “established”

Line 313: Add a comma after “MIC values”

Line 316: Add a comma after “ciprofloxacin” and “norfloxacin”

Line 320: Add a comma after “future” and insert a hyphen between “high” and “risk”

Lines 321-326: Re-write these sentences “A study conducted in Kenya between 2002 and 2009, revealed that, both MIC50 and MIC90 were highly increased (>5.0µg/ml) to penicillin and tetracycline, respectively, indicating that these antibiotics were no longer effective against N. gonorrhoeae, however, a mean MIC from 0.004 to 4.0 µg/mL for both MIC50 and MIC90 to ciprofloxacin was observed between 2007 and 2009 indicating that ciprofloxacin was since then ineffective for treating gonorrhea [3]” as “A study conducted in Kenya between 2002 and 2009 revealed that MIC50 and MIC90 were highly increased (>5.0µg/ml) to penicillin and tetracycline, respectively, indicating that these antibiotics were no longer effective against N. gonorrhoeae. However, a mean MIC from 0.004 to 4.0 µg/mL for both MIC50 and MIC90 to ciprofloxacin was observed between 2007 and 2009, indicating that ciprofloxacin was ineffective for treating gonorrhea [3].”

Lines 331-339: Re-write these sentences “The current report therefore highlights that, heterogeneous phenotypic groups of QRNG+ PPNG+ and QRNG+PPNG+TRNG+ were predominant and thus suggest a selective pressure from antibiotic use and this is a strong driver of potential emergency of AMR to currently recommended cephalosporins. A previous study conducted in 2009 - 2013 showed >10% of predominant groups of QRNG+ PPNG+ and QRNG+PPNG+TRNG+ were also observed among the isolates with HLR to ciprofloxacin[1]. The high prevalence rate of 42.7% for tetracycline (TRNG) was observed among the isolates with HLR to ciprofloxacin, although, some previous studies suggested returning to the use of ‘older’ antibiotics may be another viable option for gonorrhea treatment” as “Therefore, the current report highlights that heterogeneous phenotypic groups of QRNG+PPNG+ and QRNG+PPNG+TRNG+ were predominant and thus suggest a selective pressure from antibiotic use. This is a strong driver of the potential emergency of AMR to currently recommended cephalosporins. A previous study conducted in 2009 - 2013 showed >10% of predominant groups of QRNG+ PPNG+ and QRNG+PPNG+TRNG+ were also observed among the isolates with HLR to ciprofloxacin[1]. A high prevalence of 42.7% for tetracycline (TRNG) was observed among the isolates with HLR to ciprofloxacin, although some previous studies suggested returning to ‘older’ antibiotics may be another viable option for gonorrhea treatment.”

Lines 339-342: Re-write this sentence “Tetracycline is no longer in use in most resource-contrained countries for more than a decade, however, current reports demonstrated high prevalence of resistant N. gonorrhoeae isolates to tetracycline [1,3,7,9– 341 11].” as “Tetracycline is no longer used in most resource-constrained countries for over a decade. However, current reports demonstrated a high prevalence of resistant N. gonorrhoeae isolates to tetracycline [1,3,7,9– 341 11].”

Line 353: Replace “in the absence of” with “without”. Include “A” before “recent”

Line 355: Start a new sentence with “Increased” and add “a” before “faster”

Line 356: Replace this phrase “include but not limited to” with “include but are not limited to the”

Lines 358-360: Revise this sentence “The prevailing gonococcal strains and their antibiotic profiles can change very rapidly and therefore short-term studies need to be repeated regularly to be useful in guiding treatment regimens [23].” as “The dominant gonococcal strains and their antibiotic profiles can change rapidly, and therefore short-term studies need to be repeated regularly to guide treatment regimens [23].”

Conclusion

Line 362: Delete “The” are the beginning of the line

Lines 366-373: Revise these sentences “With evidence from other similar studies conducted in representative populations, ciprofloxacin and or norfloxacin should not be recommended for treatment of gonorrhea in the country. Despite that, low rate of resistance to ceftriaxone, cefixime, azithromycin and spectinomycin has been reported in this study, this is a critical sign that the effectiveness of current treatment options will be challenged soon. The combination of ceftriaxone and azithromycin have been recommended by other studies to be the hallmark for the treatment of gonorrhea [24].” as “With evidence from other similar studies conducted in representative populations, ciprofloxacin and norfloxacin should not be recommended to treat gonorrhoea in the country. Despite that a low rate of resistance to ceftriaxone, cefixime, azithromycin, and spectinomycin has been reported in this study, this is a critical sign that the current treatment options' effectiveness be challenged soon. Other studies have recommended the combination of ceftriaxone and azithromycin to be the hallmark for the treatment of gonorrhoea [24].”

Line 375: Re-write “are required to” as “must”

Line 376: Replace “growing” with “growth”

Lines 377-381: Replace these sentences “In vitro susceptibility studies are not always done particularly in limited resource-countries because most laboratories do not have capacity to perform microbiological methods for detection and monitoring of AMR. It is therefore further recommended that initiating and implementing surveillance programs for detection and monitoring AMR to clinical isolates in the country is imperative.” with “In vitro susceptibility studies are not always done, particularly in limited-resource countries, because most laboratories cannot perform microbiological methods to detect and monitor AMR. Therefore, it is further recommended that initiating and implementing surveillance programs for detecting and tracking AMR in clinical isolates in the country is imperative.”

Strengths

Line 385: Add “the” before “high”

Lines 386: Replace “previous” and “current” with “previously” and “currently”. Add a hyphen between “high” and “level”

Lines 388-389: Revise this phrase “The strength of the current study is the facts” as “The current study's strength is”

Line 392: Add a comma after “analysis

Author Response

MUHIMBILI UNIVERSITY OF HEALTH AND ALLIED SCIENCES

SCHOOL OF MEDICINE

       DEPARTMENT OF MICROBIOLOGY AND IMMUNOLOGY

Ref No. PF/9071/2                                                                                                                15/05/2022

Editor,

MDPI

Tropical Medicine Editorial Office

St Alban-Anlage 66, 4052 Basel

Switzerland

Re: Submission of Responses to Reviewer’s Comments on the Manuscript Titled

Antimicrobial Susceptibility Pattern and Phenotypic Analysis of Neisseria gonorrhoeae from Patients Attending Sexually Transmitted Infections Clinics in Six Regions in Tanzania” by Aboud et al 

We would like to thank reviewer 2 for very good comments and suggestions that has helped to improve the manuscript. Below are point-to-point responses to reviewer’s comments. Changes made in the revised manuscript are highlighted in yellow in the revised manuscript.

Response to Reviewer 2 Comments

Point 1: The title, as presented, captures adequately the central theme of the study. However, the authors should italicize “Neisseria gonorrhoeae” in the title.

Response 1: We concur with reviewer’s comments and Neisseria gonorrhoeae has been italicized in the title

Abstract

Point 2:

The abstract, though written to capture the objective, methods, results, and conclusion of the research lacked a good introduction/background. Furthermore, abbreviations such as PPNG, TRNG, QRNG, HLR, MDR-NG, XDR-NG, MIC, and AMR were used in the abstract but were not defined at first mention. The presentation of the abstract should be improved upon.

Response 2: We concur with reviewer’s comments. Introduction/background has been improved. Abbreviations such as PPNG, TRNG, QRNG, HLR, MDR-NG, XDR-NG, MIC, and AMR have been defined.

Other comments on the abstract

Point 3:

Line 21: “WHICH TYPE OF SPECIMENS” appears vague. What does this denote?

Response 3: urethral discharge and endocervical swabs have been added

Point 4:

 Line 22: Add “The” before “Prevalence” and a comma after “penicillin”

Response 4: “The” has been added before “Prevalence” and a comma after “penicillin”

Point 5:

Line 24: Add a comma after “73.7%” before “with”.

Response 5:  A comma has been added after “73.7%” before “with”.

Point 6:

Line 25: Add a comma between “34%” and “and”. Also ensure that “34%” is presented in the same decimal place as other presented percentages.

Response 6:  A comma has been added between “34%” and “and”. Also “34%” has been presented one decimal place as other presented percentages.

Point 7:

Line 26: Add a comma between “79.9%” and “and”.

Response 7: A comma has been added between “79.9” and “and”

Point 8:

Line 27: Include “that” after “showed”.

Response 8: “that” has been included after “showed”.

Point 9:

Line 28: Replace the comma before “associated” with “are” and include “a” before “high”.

Response 9: A comma has been replaced with “are” before “associated” and “a” has been included before “high”.

Point 10:

Line 30: Include “a” before “high prevalence”.

Response 10:  “a”  has been included before “high prevalence”.

Point 11

Line 31: Include “the” before “treatment”.

Response 11: “the” has been included before “treatment”.

Point 12:

 Line 32: include a comma before “and XDR-NG”.

Response 7: A comma has been added between reported and “and XDR-NG”.

Point 13:

Lines 31 - 36: For a better clarity, authors should modify these sentences “MDR-NG was highly reported and XDR-NG was found gradually increasing to the currently recommended cephalosporins including ceftriaxone and cefixime. Heterogeneous groups of QRNG+PPNG+ and QRNG+PPNG+TRNG were highly observed resistant to penicillin, tetracycline, ciprofloxacin and or norfloxacin. Surveillance program is imperative in the country in order to curb the spread of the AMR.” to “MDR-NG was highly reported, and XDR-NG gradually increased to the currently recommended cephalosporins, including ceftriaxone and cefixime. Heterogeneous groups of QRNG+PPNG+ and QRNG+PPNG+TRNG were highly resistant to penicillin, tetracycline, ciprofloxacin, and norfloxacin. A surveillance program is imperative in the country to curb the spread of AMR.”

Response 13: The sentence has been incorporated as per reviewers comments.

Keywords

Point 14: Are abbreviations allowed in the keywords?

Response 14: We concur with reviewers comments, all abbreviations have been stated in the long form in the key words

Introduction

The introduction, presented, is comprehensive. It addresses the import of this research. The actual problem necessitating the conduct of this research was presented and well-justified. The aim of the study was also presented. However, the introduction is flawed with several grammatical, spelling, punctuation, and presentation errors that must be fix. Some of these have been listed below.

Point 15

Line 42: For a more concise sentence, please delete “classes of”, add a comma before “including” and delete the semicolon after “including”. Also add a comma after “tetracyclines”.

Response 15: “classes of”, has been deleted add a comma has been added before “including” and semicolon after “including” has been deleted. Also, comma has been added after “tetracyclines”.

Point 16

Line 44: Replace the comma after [4] with a full stop and begin the next sentence with “Hence”.

Response 16: A comma has been replaced after [4] with a full stop and sentence has begun with “Hence”.

Point 17

Line 45: Add “to be” between “reported” and “higher”.

Response 17: Reviewers comment has been incorporated

Point 18

Line 50: Add a comma after “settings”

Response 18: A comma has been added after “settings”

Point 19

Line 52: Please change “which is able to” to “that can”

Response 19: ”which is able to” has been changed to “that can”

Point 20

Lines 53-54: Add a hyphen between “non” and “pathogenic” and delete the comma after “species”

Response 20: A hyphen has been added between “non” and “pathogenic” and the comma after “species” has been deleted

Point 21

Line 58: Add “s” to “antibiotic” and remove the comma after “and”

Response 21: “s” has been added to “antibiotic” and  comma has been removed after “and”

Point 22

Line 60: Add a comma after “cell”

Response 22: A comma has been added after “cell”

Point 23

Line 63: Replace “against” with “to”

Response 23: “against” has been replaced with “to”

Point 24

Lines 62-65: Re-write this “These mechanisms have made N. gonorrhoeae to become resistant against many classes of antimicrobial agents such as fluoroquinolones and the current recommended extended spectrum cephalosporins (ESCs) with which have undoubtedly helped it persist in the human populations [1,2,14].” sentence as “These mechanisms have made N. gonorrhoeae resistant to many antimicrobial agents such as fluoroquinolones and the current recommended extended-spectrum cephalosporins (ESCs), which have undoubtedly helped it persist in the human populations [1,2,14].”

Response 24: Reviewers comments have been incorporated.

Point 25

Line 66: Please delete “emerges and” from the sentence and add a hyphen between “plasmid” and “mediated”

Response 25: “emerges and” was deleted from the sentence and a hyphen was added between “plasmid” and “mediated”

Point 26

Line 68: Replace “previous” with “previously”

Response 26: “previous” has been replaced with “previously”

Point 27

Line 75: Replace “where there is” with “with” and add a comma after “sector”

Response 27: “where there is” has been replaced with “with” and a comma has been added after “sector”

Point 28

Line 78: Add a comma after “1977” and “1980s”. Also insert “the” before “1980s”

Response 28: A comma has been added after “1977” and “1980s”. Also “the” has been inserted before “1980s”

Point 29

Line 80: Add “a” before “high”

Response 29: “a” has been added before “high”

Point 30

Line 81: Add a comma after “2007”

Response 30: A comma has been added after “2007

Point 31

Line 83: Add a comma after “antibiotics”

Response 31: A comma has been added after “antibiotics”

Point 32

Line 85: Add a comma after “sectors”

Response 32: A comma has been added after “sectors”

Point 33

Line 87: Begin the sentence with “Recent reports” and delete “an emergence of”

Response 33: The sentence has started with “Recent reports” and “an emergence of” has been deleted

Point 34

Line 88: Replace “extended spectrum cephalosporins (ESCs)” with “ESCs” since it has been defined earlier.

Response 34: “extended spectrum cephalosporins (ESCs)” has been replaced with “ESCs

Point 35

Lines 90-93: Re-write this sentence “In most settings, ceftriaxone has been recommended as the only empirical first-line antimicrobial monotherapy for gonorrhoea [6,14,29], despite the fact that, sporadic treatment failures to ceftriaxone have been reported by WHO and CDC [6,17,33].” as “In most settings, ceftriaxone has been recommended as the only empirical first-line antimicrobial monotherapy for gonorrhoea [6,14,29], although WHO and CDC [6,17,33] have reported sporadic treatment failures with ceftriaxone.”

Response 35: Reviewers comments have been incorporated.

Point 36

Line 93: Write out “WHO” and “CDC” in full.

Response 36: “WHO” and “CDC” has been written in full

Point 37

Line 100-102: Re-write this sentence “The quality data on antimicrobial resistance is sparse or unavailable, and most treatment guidelines particularly in African countries still recommend ciprofloxacin as the first line therapy for syndromic management of gonorrhoea and other treatable STIs [22,34].” as ‘’The quality data on antimicrobial resistance is sparse or unavailable. Most treatment guidelines, particularly in African countries, still recommend ciprofloxacin as the first-line therapy for the syndromic management of gonorrhoea and other treatable STIs [22,34].”

Response 37: Reviewers comments have been incorporated

Point 38

Line 110: Re-write “first line” as “first-line” and include a comma after “gonorrhoea”

Response 38:  “first line” has been re-written as “first-line” and a comma has been included after “gonorrhoea”

Point 39

Line 112: Delete “there was”

Response 39: “there was” has been deleted

Point 40

Line 113: Replace the semi-colon after “countries” with a comma

Response 40: the semi-colon after “countries” has been replaced with a comma

Point 41

Lines 118-123: Re-write these sentences “The study aimed to determine the prevalence of AMR of N. gonorrhoeae isolates to previous and current recommended antimicrobials and characterization of phenotypic characteristics based on the combination of antimicrobial resistance mechanisms. Data generated will be used to revise current treatment guidelines and to inform policy makers for evidence based decision making for management and control of AMR of N. gonorrhoeae isolates in the country and globally.” as “The study aimed to determine the prevalence of AMR of N. gonorrhoeae isolates to previously and currently recommended antimicrobials and characterize phenotypic characteristics based on the combination of antimicrobial resistance mechanisms. Data generated will be used to revise current treatment guidelines and inform policymakers for evidence-based decision making for the management and control of AMR of N. gonorrhoeae isolates in the country and globally.”

Response 41 : Reviewers comment have been incorporated

Materials and methods

Point 42

Line 126: Change “in the study” to “in this study”

Response 42: :in the study” has been changed to “in this study”

Point 43

Line 129: Write STI/OPD in full at first mention

Response 43: STI/OPD was written in full at first mention in the abstract line 19

Point 44

Line 130: Add a comma after “Mbeya”

Response 44: A comma has been added after “Mbeya”

Point 45

Lines 129-130: How many STI/OPD were randomly selected from a total of how many clinics in each or from the six region(s)?

Response 45: Eight STI/OPD were randomly selected from a total of 16 clinics in the six regions.

Point 46

Line 131-132: For clarity, re-write this sentence “Participants were recruited and enrolled in each study site after obtaining written informed consents” as “After obtaining written informed consent, participants were recruited and enrolled in each study site.”

Response 46: the sentence has been re-written as recommended by the reviewer

Point 47

Line 133: For conciseness, replace “whose age was” with “aged” and state the exact age of the patient (not < 16 yrs old) since it’s just one patient.

Response 47: Reviewer’s comments have been incorporated

Point 48

Lines 134-141: The isolation of N. gonorrhoeae was carried out according to which protocol? Authors should cite the standard/published protocol used in the isolation

Response 48: Reviewer’s comments have been incorporated

Point 49

Line 135: Add a comma after “(MTM plate)”

Response 49: A comma has been added after “(MTM plate)”

Point 50

Line 136: “VCAT” was not defined

Response 50: VCAT has been defined

Point 51

Line 139: Re-write this phrase “a plain MTM agar plates” as “plain MTM agar plates”

Response 51: the sentence has been rephrased

Point 52

Line 141: Why have the authors decided to store the duplicates of the isolates at two different temperatures: -80 and -150 °C?

Response 52: Reviewer’s comments have been incorporated

Point 53

Line 143: The E-Test was carried out according to which protocol?

Response 53: Reviewer’s comments have been incorporated

Point 54

Line 147: Add “a” before “DST” and “the” before “quantitative”. What is DST? This abbreviation has not been defined.

Response 54: Reviewer’s comments have been incorporated

Point 55

Line 150: Add “a” before “graduated”

Response 55: “a” has been added before “graduated”

Point 56

Line 151: Include a comma after “agent”

Response 56: A comma has been included after agent

Point 57

Line 154: Re-write “a 90-mm plate,” as “a 90 mm plate”

Response 57:  a 90-mm plate,” has been re-written as “a 90 mm plate”

Point 58

Line 158: Add a comma after “plate”

Response 58: A comma has been added after “plate”

Point 59

Line 162: Add “the” before “growth”

Response 59: “the” has been added before “growth

Point 60

Line 168: Change “as per” to “based on” and add “the” before “manufacturer’s”. Add a comma after “read”.

Response 60: as per” has been changed to “based on” and  “the” has been added before “manufacturer’s”. Also a comma has been added after “read”.

Point 61

Line 169: Add “the” before “passing”

Response 61: “the” has been added before “passing

Point 62

Line 171: Add a comma after “dilution”

Response 62: A comma has been added after “dilution”

Point 63

Line 172-174: Replace this sentence “Poor or no growth in the control plate was resubcultured and repeated to avoid erroneous reading which can be a result of poor or slow growth of an isolate.” with “Poor or no growth in the control plate was subcultured and repeated to avoid erroneous readings resulting from an isolate's poor or slow growth.”

Response 63: The reviewer’s comments have been incorporated

Point 64

Line 174: Add a comma after “recorded”

Response 64: A comma has been added after “recorded”

Point 65

Line 182: Delete “individual”

Response 65: “individual” has been deleted

Point 66

Line 185: Add “were” before “inhibited”

Response 66: “were” has been added before “inhibited

Point 67

Line 187: “Strains” should not be italicized

Response 67: “Strains” has not been italicized

Point 68

Line 188: Replace “the construction of” with “constructing”

Response 68: “the construction of” has been replaced with “constructing”

Point 69

Line 195: Add a hyphen between quinolone and resistant

Response 69: A hyphen has been added between quinolone and resistant

Point 70

Line 193-195: Please re-write this sentence “. Clustering was done based on in vitro susceptibility of an isolate to ciprofloxacin either susceptible denoted as non-quinolone resistant N. gonorrhoeae (NQRNG) or quinolone resistant N. gonorrhoeae (QRNG) respectively” to make better meaning.

Response 70: the sentence has been re-written as recommended

Point 71

Line 199: Add a comma after “tetracycline”

Response 71: A comma has been added after “tetracycline”

Point 72

Line 200: Add “the” before “currently”

Response 72: “the” has been added before “currently

Point 73

Line 204: Include a hyphen between “p” and “value”. Also, the general rule is to consider significant values at p-values less than 0.05 and not less or equal to 0.05. The authors should revise their results to ensure that p-values of 0.05 were not taken as significant.

Response 73: A hyphen has been included between “p” and “value”. Also, the statistical significance was accepted when p-values were less than 0.05 not equal to 0.05.

Point 74

Line 209: Add “the” before “agar”

Response 74: “the” has been added before “agar

Point 75

Lines 212-213: Add a comma after “agents”, “cefixime’, “ceftriaxone”, and “1.2”. Also include “%” to “2.1” and “1.2”. Change the “were” before “3.5%” to “was”

Response 75: A comma has been added after “agents”, “cefixime’, “ceftriaxone”, and “1.2”. Also “%” has been included to “2.1” and “1.2”.  The word “were” before “3.5%” has been changed to “was”

Point 76

Line 215: Add a comma after “Mwanza”

Response 76: A comma has been added after “Mwanza”

Point 77

Line 217: Delete “respectively” and add a comma after “Mbeya”

Response 77:  “respectively” has been deleted and a comma has been added after “Mbeya”

Point 78

Line 218: Add “the” before “Mwanza region”

Response 78: “the” has been added before “Mwanza region

Point 79

Line 219: Add a comma after “10.5%” and delete “both” before “PPNG”

Response 79: A comma has been added after “10.5%” and d “both” has been deleted before “PPNG”

Point 80

Line 220-221: Add a comma after “4.2%” and “Dar es Salaam”

Response 80: A comma has been added after “4.2%” and “Dar es Salaam”

Point 81

Line 222: Delete “both” before “Mbeya”

Response 81: Both has been deleted before “Mbeya”

Point 82

Line 223: Add a comma after “Mwanza”

Response 82: A comma has been added after “Mwanza”

Point 83

Lines 226-227: Fix the punctuation errors in this sentence “MIC50 and MIC90 of ciprofloxacin and norfloxacin to N. gonorrhoeae were 4.0 and 8.0 and, 2.0 and 4.0 µg/mL respectively” by re-writing it as “MIC50 and MIC90 of ciprofloxacin and norfloxacin to N. gonorrhoeae were 4.0 and 8.0, and 2.0 and 4.0 µg/mL, respectively.”

Response 83: The punctuation error has been fixed

Point 84

Line 227-228: For better clarity, re-write this sentence “Mean MICs of 0.023, 0.047 and 0.064µg/mL of ceftriaxone, cefixime and cefuroxime were observed in the study” as “The study observed the mean MICs of 0.023, 0.047, and 0.064 µg/mL of ceftriaxone, cefixime, and cefuroxime.”

Response 84: The reviewer’s comments have been incorporated

Point 85

Line 231: Add “a” before “mean”

Response 85: “a” has been added before “mean”

Point 86

Line 237: Begin the sentence with “The”

Response 86: the sentence has begun with “The”

Point 87

Line 238: Add a comma before “with”

Response 87: A comma has been added before “with

Point 88

Line 240: Add a comma after “(38)”

Response 7: A comma was added after “38”

Point 89

Line 243: Add a comma after “groups” and delete “which were”

Response 89: A comma was added after groups and “which were” was deleted

Point 90

Line 244: Add “the” before “euclidean”

Response 90: “the” was added before “euclidean”

Point 91

Line 245: Add “a” before “dendogram”

Response 91: “a” was added before “dendogram”

Point 92

Line 246: Add “and” before “had” and delete the comma after “cluster”. Add a comma after “LLR to quinolone”

Response 92: “and” was added before “had” and  the comma was deleted after “cluster”. Also, a comma was added after “LLR to quinolone”

Point 93

Line 249: Add a comma after “isolates” and include “the” after “by”

Response 93: A comma was added after “isolates” and “the” was included after “by”

Point 94

Line 253-254: Rephrase “high AMR to” to “higher AMR than”

Response 94: “high AMR to” was rephrased to “higher AMR than”

Point 95

Line 258: Insert a hyphen between “penicillin” and “sensitive”

Response 95: A hyphen was inserted between “penicillin” and “sensitive”

Point 96

Line 259: Delete “that were”

Response 96:  “that were” was deleted

Discussion

The results were well discussed. However, the authors must fix all grammatical, punctuation, spelling, and presentation errors noticed.

Point 97

Line 281: Add “a” before “high”

Response 97: “a” was added before “high

Point 98

Line 282: Add a comma after “agents”

Response 98: A comma was added after “agents”

Point 99

Lines 282-285: Modify this sentence “Treatment failure for gonorrhoea by ciprofloxacin has been reported by many previous studies, despite the fact that, in resource limited countries, ciprofloxacin is still used to treat gonorrhoea and other infections, especially in informal health sectors.” as “Many previous studies have reported treatment failure for gonorrhoea by ciprofloxacin, and even though, in resource-limited countries, ciprofloxacin is still used to treat gonorrhoea and other infections, especially in informal health sectors.”

Response 99:  The reviewer’s comment has been incorporated

Point 100

Line 287: Delete “both” after “for”

Response 100:  “both” was deleted after “for”

Point 101

Line 291: Delete “as” after “definitions”

Response 101:  “as” was deleted after “definitions

Point 102

Lines 294-296: Re-present this sentence “In African countries, penicillin resistance was 65%, tetracycline 97%, and ciprofloxacin 11% as reported in Kisumu, Kenya from 2002 and 2009[3].” as “In African countries, penicillin, tetracycline, and ciprofloxacin resistance was 65%, 97%, and 11%, as reported in Kisumu, Kenya, from 2002 to 2009[3].”

Response 102:  Reviewer’s comments was incorporated

Point 103

Line 298: Change the “The” at the beginning of the line to “An”, add a comma after “2012” and delete the “in” after “2012”

Response 103:  “the” was changed to “An” and a comma was added after “2012”

Point 104

Line 302: Delete the “resistance” after “vitro”

Response 104:  “resistance” was deleted after “vitro”

Point 105

Line 303: Add “a” before “case by case” and insert a hyphen between “cross” and “sectional”

Response 105:  ““a” was added before “case by case” and a hyphen was inserted between “cross” and “sectional”

Point 106

Line 306-309: Re-write this sentence “In Tanzania cephalosporins such as ceftriaxone, cefixime and cefuroxime and macrolides (azithromycin and erythromycin) are frequently used both in formal and informal health sectors and sometimes are found in over the counter (OTC) shops” as “In Tanzania, cephalosporins such as ceftriaxone, cefixime, cefuroxime, and macrolides (azithromycin and erythromycin) are frequently used in formal and informal health sectors and sometimes are found in over the counter (OTC) shops.”

Response 106:  The reviewer’s comment has been incorporated

Point 107

Line 311: Add a comma after “established”

Response 107:  a comma was added after “established”

Point 108

Line 313: Add a comma after “MIC values”

Response 108:  a comma was added after “MIC values”

Point 109

Line 316: Add a comma after “ciprofloxacin” and “norfloxacin”

Response 109:  a comma was added after “ciprofloxacin” and “norfloxacin”

Point 110

Line 320: Add a comma after “future” and insert a hyphen between “high” and “risk”

Response 110:  a comma was added after “future” and a hyphen inserted between “high” and “risk”

Point 111

Lines 321-326: Re-write these sentences “A study conducted in Kenya between 2002 and 2009, revealed that, both MIC50 and MIC90 were highly increased (>5.0µg/ml) to penicillin and tetracycline, respectively, indicating that these antibiotics were no longer effective against N. gonorrhoeae, however, a mean MIC from 0.004 to 4.0 µg/mL for both MIC50 and MIC90 to ciprofloxacin was observed between 2007 and 2009 indicating that ciprofloxacin was since then ineffective for treating gonorrhea [3]” as “A study conducted in Kenya between 2002 and 2009 revealed that MIC50 and MIC90 were highly increased (>5.0µg/ml) to penicillin and tetracycline, respectively, indicating that these antibiotics were no longer effective against N. gonorrhoeae. However, a mean MIC from 0.004 to 4.0 µg/mL for both MIC50 and MIC90 to ciprofloxacin was observed between 2007 and 2009, indicating that ciprofloxacin was ineffective for treating gonorrhea [3].”

Response 111:  The reviewer’s comments has been incorporated

Point 112

Lines 331-339: Re-write these sentences “The current report therefore highlights that, heterogeneous phenotypic groups of QRNG+ PPNG+ and QRNG+PPNG+TRNG+ were predominant and thus suggest a selective pressure from antibiotic use and this is a strong driver of potential emergency of AMR to currently recommended cephalosporins. A previous study conducted in 2009 - 2013 showed >10% of predominant groups of QRNG+ PPNG+ and QRNG+PPNG+TRNG+ were also observed among the isolates with HLR to ciprofloxacin[1]. The high prevalence rate of 42.7% for tetracycline (TRNG) was observed among the isolates with HLR to ciprofloxacin, although, some previous studies suggested returning to the use of ‘older’ antibiotics may be another viable option for gonorrhea treatment” as “Therefore, the current report highlights that heterogeneous phenotypic groups of QRNG+PPNG+ and QRNG+PPNG+TRNG+ were predominant and thus suggest a selective pressure from antibiotic use. This is a strong driver of the potential emergency of AMR to currently recommended cephalosporins. A previous study conducted in 2009 - 2013 showed >10% of predominant groups of QRNG+ PPNG+ and QRNG+PPNG+TRNG+ were also observed among the isolates with HLR to ciprofloxacin[1]. A high prevalence of 42.7% for tetracycline (TRNG) was observed among the isolates with HLR to ciprofloxacin, although some previous studies suggested returning to ‘older’ antibiotics may be another viable option for gonorrhea treatment.”

Response 112:  The reviewer’s comments have been incorporated

Point 113

Lines 339-342: Re-write this sentence “Tetracycline is no longer in use in most resource-constrained countries for more than a decade, however, current reports demonstrated high prevalence of resistant N. gonorrhoeae isolates to tetracycline [1,3,7,9– 341 11].” as “Tetracycline is no longer used in most resource-constrained countries for over a decade. However, current reports demonstrated a high prevalence of resistant N. gonorrhoeae isolates to tetracycline [1,3,7,9– 341 11].”

Response 113:  The reviewer’s comments have been incorporated

Point 114

Line 353: Replace “in the absence of” with “without”. Include “A” before “recent”

Response 114:  “in the absence of” has been replaced with “without”.  “A” was included before “recent”

Point 115

Line 355: Start a new sentence with “Increased” and add “a” before “faster”

Response 115:  A new sentence was started with “Increased” and “a” has been added before “faster”

Point 116

Line 356: Replace this phrase “include but not limited to” with “include but are not limited to the”

Response 116:  The phrase has been replaced as per reviewer’s comment

Point 117

Lines 358-360: Revise this sentence “The prevailing gonococcal strains and their antibiotic profiles can change very rapidly and therefore short-term studies need to be repeated regularly to be useful in guiding treatment regimens [23].” as “The dominant gonococcal strains and their antibiotic profiles can change rapidly, and therefore short-term studies need to be repeated regularly to guide treatment regimens [23].”

Response 117:  The reviewer’s comment has been incorporated

Conclusion

Point 118

Line 362: Delete “The” at the beginning of the line

Response 118: “The” was deleted at the beginning of the line

Point 119

Lines 366-373: Revise these sentences “With evidence from other similar studies conducted in representative populations, ciprofloxacin and or norfloxacin should not be recommended for treatment of gonorrhea in the country. Despite that, low rate of resistance to ceftriaxone, cefixime, azithromycin and spectinomycin has been reported in this study, this is a critical sign that the effectiveness of current treatment options will be challenged soon. The combination of ceftriaxone and azithromycin have been recommended by other studies to be the hallmark for the treatment of gonorrhea [24].” as “With evidence from other similar studies conducted in representative populations, ciprofloxacin and norfloxacin should not be recommended to treat gonorrhoea in the country. Despite that a low rate of resistance to ceftriaxone, cefixime, azithromycin, and spectinomycin has been reported in this study, this is a critical sign that the current treatment options' effectiveness be challenged soon. Other studies have recommended the combination of ceftriaxone and azithromycin to be the hallmark for the treatment of gonorrhoea [24].”

Response 119:  The reviewer’s comment has been incorporated

Point 120

Line 375: Re-write “are required to” as “must”

Response 120:  “are required to” was re-written as “must”

Point 121

Line 376: Replace “growing” with “growth”

Response 121:  “growing” was replaced with “growth”

Point 122

Lines 377-381: Replace these sentences “In vitro susceptibility studies are not always done particularly in limited resource-countries because most laboratories do not have capacity to perform microbiological methods for detection and monitoring of AMR. It is therefore further recommended that initiating and implementing surveillance programs for detection and monitoring AMR to clinical isolates in the country is imperative.” with “In vitro susceptibility studies are not always done, particularly in limited-resource countries, because most laboratories cannot perform microbiological methods to detect and monitor AMR. Therefore, it is further recommended that initiating and implementing surveillance programs for detecting and tracking AMR in clinical isolates in the country is imperative.”

Response 122:  The sentence has been replaced as suggested by the reviewer

Strengths

Point 123

Line 385: Add “the” before “high”

Response 123:  “the” has been added before “high”

Point 124

Lines 386: Replace “previous” and “current” with “previously” and “currently”. Add a hyphen between “high” and “level”

Response 124:  previous” and “current” has been replaced with “previously” and “currently”. A hyphen has been added between “high” and “level”

Point 125

Lines 388-389: Revise this phrase “The strength of the current study is the facts” as “The current study's strength is”

Response 125:  The phrase has been revised

Point 126

Line 392: Add a comma after “analysis

Response 126:  a comma was added after “analysis”

We look forward to hearing from you soon

Said Aboud

Reviewer 3 Report

Specific comments:

  1. Abstract: Please perform a review of the abstract, mostly in the abreviations used. Be aware that we only use an abbreviation after the complete word is presented.
  2. Introduction section:
    1. It is necessary to include several references in this section that supports sentences (lines 54, 56 and 59).
    2. In lines 80 to 86 please rewrite the sentence, it is confuse for the reader.

  1. Materials and methods section:
    1. It is necessary to include several references in this section that supports sentences (lines 136, 138, 141, 150, 156, 168, 197 and 203).
    2. Did you confirm the isolates identity using molecular or biochemical methods? This should be included, to guarantee the isolates species.
    3. In line 159-160 you mention “…a standardized inoculum of bacterial suspension…”. What do you mean by that? Please clarify this information
    4. All the Antimicrobial susceptibility testing should follow standard procedures like CLSI or Eucast methods. Please include the references and standard methods used all over this section.
    5. The classification of MDR or XDR isolates should be based in standard methods or references. Please include the references used for these classifications.

  1. Results section:
    1. In several points of these section appear some Upper letters that does not make sense. Please clarify and correct this point (Lines 210, 211, 215, 216 and 257) For example “…tetracycline AA (98.6%)..”, “…followed by ZZ (9%)…” or even “…were susceptible to WHAT?...”
    2. In line 237 “Overall prevalence of ßeta-lactamase producing isolates (PPNG)…” how can you say that you have ßeta-lactamase producing isolates? You have not evaluated the presence of these enzymes or make any molecular evaluation. Please clarify this.
    3. In line 241 “Prevalence of plasmid-mediated tetracycline resistant N. gonorrhoeae (TRNG), non-TRNG and tetracycline-susceptible were 89.3% (125), 9.3% (13) and 1.4% (2), respectively.” Did you evaluate the presence of plasmids? How did you perform it? Clarify this point.
    4. In Table 1 the description must be improved. Please include the references used (CLSI, EUCAST, …) the method used to evaluate the isolates.
    5. In Table 2 please include the number of isolates. Also, you never mention the difference and number of isolates obtained from Men and Woman. This must be included in the results section and not only in this table.
  2. In the Discussion section:
    1. Include references in lines 285, 309, 315, 321, 334, 339, 344, 345, 349 and 358 .
    2. In line 300 “Five percent of NG strains….” Please write the correct name of the bacteria. As in line 330 and 348.
    3. Please explain the sentence “…slight paradigm shift to the right…” ( line 319).
    4. In lines 331-334. Why do you say that your results suggest a selective pressure from antibiotics use??
    5. In line 348-349 what do you mean by “WHO “O” reference strain…”??

Author Response

Ref No. PF/9071/3                                                                                                                15/05/2022

Editor,

MDPI

Tropical Medicine Editorial Office

St Alban-Anlage 66, 4052 Basel

Switzerland

Re: Submission of Responses to Reviewer’s Comments on the Manuscript Titled

Antimicrobial Susceptibility Pattern and Phenotypic Analysis of Neisseria gonorrhoeae from Patients Attending Sexually Transmitted Infections Clinics in Six Regions in Tanzania” by Aboud et al 

We would like to thank reviewer 3 for very good comments and suggestions that has helped to improve the manuscript. Below are point-to-point responses to reviewer’s comments. Changes made in the revised manuscript are highlighted in yellow in the revised manuscript.

Responses to Reviewer 3 Comments

Point 1

Abstract: Please perform a review of the abstract, mostly in the abreviations used. Be aware that we only use an abbreviation after the complete word is presented.

Response 1: Reviewer’s comments have been incorporated

Introduction section

Point 2

It is necessary to include several references in this section that supports sentences (lines 54, 56 and 59).

Response 2: References number 3 and 6 have been added

Point 3

In lines 80 to 86 please rewrite the sentence, it is confuse for the reader.

Response 3: Reviewer’s comments have been incorporated to make the sentence clear

Materials and methods section

Point 4

It is necessary to include several references in this section that supports sentences (lines 136, 138, 141, 150, 156, 168, 197 and 203).

Response 4: References have been added as recommended by reviewers

Point 5

Did you confirm the isolates identity using molecular or biochemical methods? This should be included, to guarantee the isolates species.

Response 5: The isolated were identified by biochemical tests and confirmed by Real time PCR assay.

Point 6

In line 159-160 you mention “…a standardized inoculum of bacterial suspension…”. What do you mean by that? Please clarify this information

Response 6: It is equivalent to 0.5 McFarland or 1.0 x 108cfu/ml of bacterial suspension; it is clarified in the text  

Point 7

All the Antimicrobial susceptibility testing should follow standard procedures like CLSI or Eucast methods. Please include the references and standard methods used all over this section.

Response 7: CLSI and EUCAST methods were used for Antimicrobial susceptibility testing and interpretations of results. The references are quoted in the text in line 181-186

Point 8

The classification of MDR or XDR isolates should be based in standard methods or references. Please include the references used for these classifications.

Response 8: References have been added

Results section

Point 9

In several points of these section appear some Upper letters that does not make sense. Please clarify and correct this point (Lines 210, 211, 215, 216 and 257) For example “…tetracycline AA (98.6%)..”, “…followed by ZZ (9%)…” or even “…were susceptible to WHAT?...”

Response 9: Reviewer’s comments have been addressed

Point 10

In line 237 “Overall prevalence of ßeta-lactamase producing isolates (PPNG)…” how can you say that you have ßeta-lactamase producing isolates? You have not evaluated the presence of these enzymes or make any molecular evaluation. Please clarify this.

Response 10: Reviewer’s comments have been addressed

Point 11

In line 241 “Prevalence of plasmid-mediated tetracycline resistant N. gonorrhoeae (TRNG), non-TRNG and tetracycline-susceptible were 89.3% (125), 9.3% (13) and 1.4% (2), respectively.” Did you evaluate the presence of plasmids? How did you perform it? Clarify this point.

Response 11: Reviewer’s comments have been addressed

Point 12

In Table 1 the description must be improved. Please include the references used (CLSI, EUCAST, …) the method used to evaluate the isolates.

Response 13: References for CLISI and EUCAST have been mentioned in the methods section in line 181-186.

Point 13

In Table 2 please include the number of isolates. Also, you never mention the difference and number of isolates obtained from Men and Woman. This must be included in the results section and not only in this table.

Response 13: Number of N gonorrhoeae isolates obtained from men and women have been included in the results section table 2

In the Discussion section

Point 14

Include references in lines 285, 309, 315, 321, 334, 339, 344, 345, 349 and 358 .

Response 8: References have been added

Point 15

In line 300 “Five percent of NG strains….” Please write the correct name of the bacteria. As in line 330 and 348.

Response 15: Reviewer’s comments have been incorporated

Point 16

Please explain the sentence “…slight paradigm shift to the right…” ( line 319).

Response 16: Reviewer’s comments have been incorporated

Point 17

In lines 331-334. Why do you say that your results suggest a selective pressure from antibiotics use??

Response 17: Selective pressure has been suggested due to possible irrational and inappropriate antibiotic use as well as self-medication for treatment of gonorrhea

Point 18

In line 348-349 what do you mean by “WHO “O” reference strain…”??

Response 18: It refers to molecular resistance of N gonorrhoeae isolates to ceftriaxone correlated to 1µg/mL

We look forward to hearing from you soon

Said Aboud

Round 2

Reviewer 1 Report

providing some data related to differential diagnosis

Author Response

Ref No. PF/9071/4                                                                                                                25/05/2022

Editor,

MDPI

Tropical Medicine Editorial Office

St Alban-Anlage 66, 4052 Basel

Switzerland

Re: Submission of Responses to Reviewer’s Comments on the Manuscript Titled

Antimicrobial Susceptibility Pattern and Phenotypic Analysis of Neisseria gonorrhoeae from Patients Attending Sexually Transmitted Infections Clinics in Six Regions in Tanzania” by Aboud et al 

We would like to thank reviewer 1 for very good comments and suggestions that has helped to improve the manuscript. Below are point-to-point responses to additional reviewer’s comments. Changes made in the revised manuscript are highlighted in yellow.

Responses to Reviewer 1 Comments

Point

(x) English language and style are fine/minor spell check required
Response

Minor spell check has been done

Point

Yes

Can be improved

Must be improved

Not applicable

Does the introduction provide sufficient background and include all relevant references?

(x)

( )

( )

( )

Response

We thank the reviewer for very good feedback

Point

Are all the cited references relevant to the research?

( )

(x)

( )

( )

Response

A total of nine new references were added in the

revised manuscript (25, 33, 34, 35, 36, 39, 41, 42, 44).

We would appreciate very much to add any additional

references if they would be specified.

Point

Is the research design appropriate?

(x)

( )

( )

( )

Response

We thank the reviewer for very good feedback

Point

Are the methods adequately described?

(x)

( )

( )

( )

Response

We thank the reviewer for very good feedback

Point

Are the results clearly presented?                 (x)

( )

( )

( )

Response

We thank the reviewer for very good feedback

Point

Are the conclusions supported by the results? (x) ( ) ( ) ( )   

Response

We thank the reviewer for very good feedback

Point: providing some data related to differential
diagnosis

Response: The primary aim of the current paper is to present data on antimicrobial susceptibility testing patterns on N. gonorrhoeae isolates.
Data on C. trachomatis, M. genitalium and G. vaginalis are presented in another paper.

Reviewer 2 Report

All comments raised in the earlier version of the manuscript have been addressed. Well done.

Author Response

Ref No. PF/9071/5                                                                                                                25/05/2022

Editor,

MDPI

Tropical Medicine Editorial Office

St Alban-Anlage 66, 4052 Basel

Switzerland

Re: Submission of Responses to Reviewer’s Comments on the Manuscript Titled

Antimicrobial Susceptibility Pattern and Phenotypic Analysis of Neisseria gonorrhoeae from Patients Attending Sexually Transmitted Infections Clinics in Six Regions in Tanzania” by Aboud et al 

We would like to thank reviewer 2 for very good comments and suggestions that has helped to improve the manuscript. Below are point-to-point responses to additional reviewer’s comments. Changes made in the revised manuscript are highlighted in yellow.

Response to Reviewer 2 Comments

Point

(x) I would like to sign my review report

Response

We thank the reviewer for feedback

Point

(x) Moderate English changes required
Response

We would appreciate very much if additional moderate English changes required would be specified

Point

Yes

Can be improved

Must be improved

Not applicable

Does the introduction provide sufficient background and include all relevant references?

(x)

( )

( )

( )

Response

We thank the reviewer for very good feedback

Point

Are all the cited references relevant to the research?

(x)

( )

( )

( )

Response

We thank the reviewer for very good feedback

Point

Is the research design appropriate?

(x)

( )

( )

( )

Response

We thank the reviewer for very good feedback

Point

Are the methods adequately described?

(x)

( )

( )

( )

Response

We thank the reviewer for very good feedback

Point

Are the results clearly presented?

(x)

( )

( )

( )

Response

We thank the reviewer for very good feedback

Point

Are the conclusions supported by the results?

(x)

( )

( )

( )

Response

We thank the reviewer for very good feedback

Reviewer 3 Report

Lines 144-145: Please refer what primers and conditions were used in the PCR protocol.

Author Response

Ref No. PF/9071/6                                                                                                                25/05/2022

Editor,

MDPI

Tropical Medicine Editorial Office

St Alban-Anlage 66, 4052 Basel

Switzerland

Re: Submission of Responses to Reviewer’s Comments on the Manuscript Titled

Antimicrobial Susceptibility Pattern and Phenotypic Analysis of Neisseria gonorrhoeae from Patients Attending Sexually Transmitted Infections Clinics in Six Regions in Tanzania” by Aboud et al 

We would like to thank reviewer 3 for very good comments and suggestions that has helped to improve the manuscript. Below are point-to-point responses to additional reviewer’s comments. Changes made in the revised manuscript are highlighted in yellow.

Responses to Reviewer 3 Comments

Point

(x) I would not like to sign my review report
Response

We thank the reviewer for feedback

Point

(x) English language and style are fine/minor spell check required

Response

Minor spell check has been done

Point

Yes

Can be improved

Must be improved

Not applicable

Does the introduction provide sufficient background and include all relevant references?

(x)

( )

( )

( )

Response

We thank the reviewer for very good feedback

Point

Are all the cited references relevant to the research?

(x)

( )

( )

( )

Response

We thank the reviewer for very good feedback

Point

Is the research design appropriate?

(x)

( )

( )

( )

Response

We thank the reviewer for very good feedback

Point

Are the methods adequately described?

(x)

( )

( )

( )

Response

We thank the reviewer for very good feedback

Point

Are the results clearly presented?

(x)

( )

( )

( )

Response

We thank the reviewer for very good feedback

Point

Are the conclusions supported by the results?

(x)

( )

( )

( )

Response

We thank the reviewer for very good feedback

Comments and Suggestions for Authors

Point

Lines 144-145: Please refer what primers and conditions were used in the PCR protocol.

Response

Artus CT/NG QS-RGQ Real Time PCR assay (QIAGEN GmbH, Germany) targeting a 86 bp region of the C. trachomatis plasmid, a 66 bp region of the C. trachomatis genome, and a 74 bp target of the N. gonorrhoeae genome was used for qualitative detection of the same. Conditions were at 37 °C for 30 minutes followed by 70 °C and 15-25 °C. 
